# Selfish mutations promote age-associated erosion of mtDNA integrity in mammals

Ekaterina Korotkevich [1] ✉, Daniel N. Conrad [2], Zev J. Gartner [2] & Patrick H. O'Farrell [1] ✉

Mutations in mitochondrial DNA (mtDNA) accumulate during aging and contribute to age-related conditions. High mtDNA copy number masks newly emerged recessive mutations; however, phenotypes develop when cellular levels of a mutant mtDNA rise above a critical threshold. The process driving this increase is unknown. Single-cell DNA sequencing of mouse and human hepatocytes detected increases in abundance of mutant alleles in sequences governing mtDNA replication. These alleles provided a replication advantage (drive) leading to accumulation of the affected genome along with a wide variety of associated passenger mutations, some of which are detrimental. The most prevalent human mtDNA disease variant, the 3243A>G allele, behaved as a driver, suggesting that drive underlies prevalence. We conclude that replicative drive amplifies linked mtDNA mutations to a threshold at which phenotypes are seen thereby promoting age-associated erosion of the mtDNA and influencing the transmission and progression of mitochondrial diseases.

Eukaryotic cells contain many copies of mtDNA. Normally, all these copies are equivalent; however, mutations arise in individual copies, creating heterogeneity or heteroplasmy[1]. To persist, a newly formed mutation must compete with the abundant copies of wild type (WT) genomes. Furthermore, since deleterious mtDNA can be masked by the co-resident WT genomes, to have an impact, a deleterious mutation must outpace WT genomes to reach a critical threshold, usually in the range of 60–90%[2]. Thus, the impact of a deleterious mtDNA mutation, whether inherited or acquired de novo during aging, will depend on mechanisms influencing the cellular abundance of the mutant mtDNA. Our interest is in the increase in abundance of mtDNA mutations emerging in somatic tissues during aging.

An adult human carries ~$10^{16}$ mtDNAs, and this huge population turns over about 1000 times[3] in a lifetime. All possible simple mutations will occur billions of times and the resulting genomes will compete in an evolutionary process within our bodies[4]. This competition is affected by genetic drift—a random shift in the ratio of mutant and WT genomes due to stochastic nature of mtDNA replication and segregation. In addition to random genetic drift, negative and positive selection influence whether mutant variants wane or thrive. In diverse organisms, the outcome of this competition is the accumulation of mtDNA mutations in somatic tissues with age[5–9].

One mode of positive selection has been described in systems from yeast to humans, where genetic and genomic approaches identified mtDNA variants that have a replicative advantage over WT mtDNA[10–15]. Often, such variants outcompete coresident WT genomes even when they have negative consequences on cellular and organismal function[10–13]. However, this replicative advantage is limited to rare mutations, usually in noncoding sequences and so does not immediately explain the large diversity of alleles, including those impacting protein function that climb to high abundance[16].

Here, we explore how mtDNA mutations increase in abundance as mice and humans age. We use the power of single-cell mtDNA sequencing combined with computational simulation to show that diverse mtDNA mutations attain high cellular abundance by physical linkage to alleles that drive positive selection. We show that such "driver-passenger" partnerships are a major route by which newly emerging mutations climb in abundance within cells.

[1]Department of Biochemistry and Biophysics, University of California, San Francisco, San Francisco, California, USA. [2]Department of Pharmaceutical Chemistry, University of California, San Francisco, San Francisco, California, USA. ✉e-mail: ekaterina.korotkevich@ucsf.edu; ofarrell@cgl.ucsf.edu

## Results

### Spectrum of mtDNA mutations

In mice, most de novo somatic mtDNA mutations are present at levels much lower than 1% of the total mtDNA in a tissue, below the detection capabilities of standard Illumina sequencing[8]. Since each base pair (bp) can mutate to any of three alternatives or be affected by indels of various sizes, the number of possible alleles in the ~16 kb mtDNA is large (>48,000). Each specific mtDNA mutation (allele) would occur in only a tiny fraction of the cells, but in these cells and their descendants, the relative abundance of the allele would be much higher than in the whole tissue and thus more readily detectable with standard next-generation sequencing methods (Fig. 1a). Therefore, we developed high-throughput sequencing methods to profile mtDNA mutations

in single hepatocytes that are similar to previously published approaches[17,18] (Fig. 1b, Supplementary Figs. 1 and 2).

To assay mtDNA sequences, we employed ATAC-seq[19]. This technique is commonly used to assess chromatin accessibility, but, with only minor modifications, can be used to generate libraries enriched in mtDNA sequences[17,18] (Supplementary Figs. 1b and 2c). We coupled this with FACS, or the 10X Genomics platform, to analyze hundreds to thousands of single cells. We further enhanced the 10X-based approach by adapting a strategy, initially developed for RNA sequencing, that uses sample specific barcodes to multiplex 10X samples[20,21] (Supplementary Fig. 2). This strategy minimizes technical variation and cost, while allowing detection of multiplets (Methods).

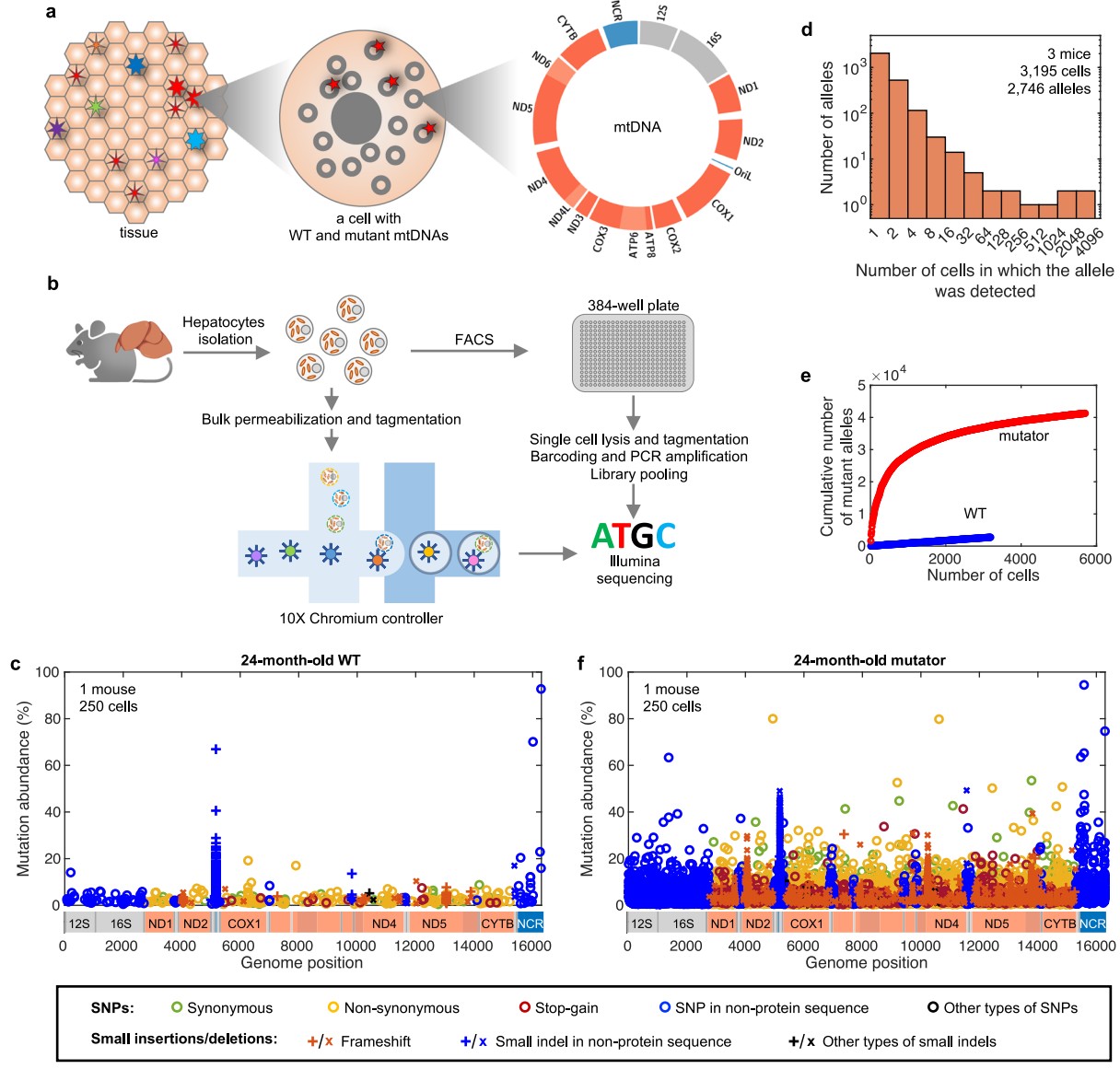

**Fig. 1 | Single-cell sequencing for profiling de novo somatic mtDNA mutations.** **a** De novo somatic mtDNA mutations occur infrequently, such that each allele is generally present in a few cells of a tissue. mtDNA map was created with *Circos*[63]. **b** Schematic of steps in plate-based and 10X-based single-cell mtDNA sequencing used to profile mtDNA mutations. Parts of the schematic were created in BioRender. Korotkevich, E. (2025) https://BioRender.com/c17m373. **c** Spectrum of mtDNA mutations in hepatocytes from a 24-month-old C57BL/6J mouse. Distinct symbols indicate allele types. Each occurrence of a mutation is represented by a symbol indicating its genomic position (*X*-axis) and its abundance (percent of reads, *Y*-axis) in the cell in which the mutation was recorded (see Supplementary

Fig. 1d for measurement precision). Data from 250 cells are aggregated in the plot. **d** Frequency of mutant allele detection in cells from three 24-month-old C57BL/6J mouse livers. Most alleles are seen in one or a few cells, but rare alleles are found in most cells. **e** The number of detected distinct mutant alleles increases with the number of cells analyzed. An analysis of 5701 cells from three 24-month-old heterozygous mutator mice detected 1,209,103 mutations representing 41,273 distinct alleles. An analysis of 3195 cells from three 24-month-old C57BL/6J WT mice detected 14,581 mutations representing 2746 alleles. **f** Spectrum of mtDNA mutations in hepatocytes from a 24-month-old heterozygous mutator mouse.

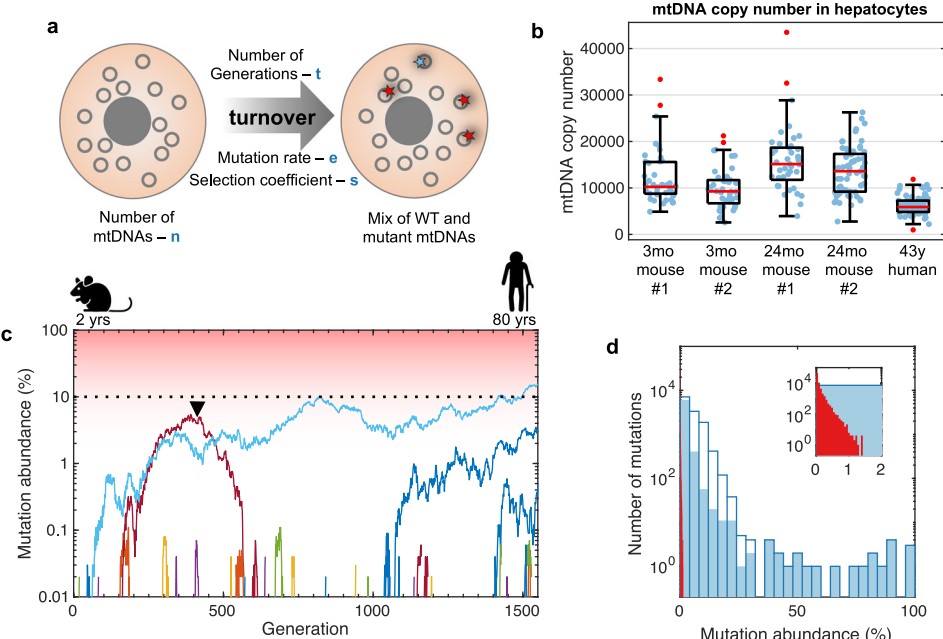

**Fig. 2 | Neutral de novo mtDNA mutations fluctuate in abundance but high-abundance mutations are infrequent, and their likelihood increases slowly with age. a** Schema of simulations of mtDNA mutation accumulation. **b** ddPCR measurements of mtDNA copy number (blue points) in single hepatocytes from young and old WT C57BL/6J mice and a middle-aged human ($n$ = 43, 44, 41, 65, and 71 cells, respectively). Boxes indicate the 25th and 75th percentiles, the red line marks the median. The whiskers extend to the most extreme data points not considered outliers (conventionally defined as values beyond 1.5 times the inter-quartile range above the upper quartile and below the lower quartile; red points). **c** Dynamics of accumulation of simulated neutral de novo somatic mutations. The plot tracks the fate of a generic allele as mutations emerge in 250 simulations of single cells over time. A total of 80 mutations (colored lines) emerged. Most disappeared shortly after emergence. Only three persisted to the end of the simulations and only one reached an abundance of 10% (black dotted line). The arrowhead marks a mutation that was lost after an initial rise. Model parameters: mutation rate 10,000 genomes/cell and $2 \times 10^{-8}$/bp/replication cycle. The mouse and human icons were created in BioRender. Korotkevich, E. (2025) https://BioRender.com/d98f984. **d** Comparison of the abundance distribution of mutant mtDNA in hepatocytes from 24-month-old mice (blue; $n$ = 3 mice, 3195 cells, 12,954 detected mutations; these data are the same as those presented in Supplementary Fig. 5) with distributions predicted by simulations of random drift (red bars on the left; 3195 simulated cells, 65,772 mutations). Open bars show data excluding a single clonal mutant allele. Filled bars also exclude indels in the homopolymeric sequence of OriL. The inset shows data with abundances below 2% to allow visualization of the simulated data, which fall entirely within the first bar of the experimental data. Model parameters: 16,299 bp genome, 10,000 genomes/cell, mutation rate $2 \times 10^{-8}$/bp/replication cycle, 40 Generations.

Figure 1c aggregates data from 250 cells (from a larger dataset) of a 24-month-old C57BL/6J mouse and displays the cellular abundance and genomic location of each detected mutation. Figure 1d shows the frequency with which distinct alleles were found in a population of 3195 hepatocytes from three 24-month-old C57BL/6J mice. Notably, most alleles (2045 of 2746) were detected only once. Consequently, as we examine more cells from such a population, the number of distinct alleles increases almost linearly (Fig. 1e). To increase the depth of analysis, we also profiled mtDNA mutations in heterozygous mutator mice that have a paternally derived allele of a proofreading-deficient mtDNA polymerase[22,23] (Fig. 1f and Supplementary Fig. 3c, d). These mice exhibit an increased frequency of mtDNA mutations but lack the recessive premature aging phenotypes seen in the homozygote. Our analysis of 5701 hepatocytes from 24-month-old heterozygous mutator mice detected over one million mutations but now many alleles are detected repeatedly (Fig. 1e).

We compared the spectrum of mtDNA mutations in livers of aged (24 months) and young (3 months) WT and heterozygous mutator mice (Fig. 1c, f and Supplementary Fig. 3a–d). The number of detected mutant mtDNA alleles increased from 3 to 24 months of age by 6-fold in WT mice (Supplementary Fig. 3). At both ages, most, but not all, alleles were present at low abundance. Among exceptionally abundant alleles were small indels in a homopolymeric sequence in the light strand origin of replication (OriL; positions 5172–5182) (Fig. 1c, f and Supplementary Fig. 3a–d) and clonally expanded mutations (which appeared sporadically in single mice, e.g., Supplementary Fig. 3a) that emerged early in development or were maternally transmitted.

Beyond these special cases, many SNPs and simple indels also reached high cellular levels in aged mice. We focused our analysis on the potentially influential alleles that reach high cellular abundance. To determine whether their high abundance represented the extremes of random genetic drift or required a specific process promoting their rise in abundance, we compared our data to simulations.

## Modeling accumulation of mutant mtDNAs

We simulated the accumulation of mtDNA mutations using SLiM[24] – a forward genetic simulation framework, capable of modeling complex evolutionary scenarios (Fig. 2a). We treated mtDNAs as if they were individuals, and all mtDNA copies in a cell as a population of constant size. mtDNA copy number was directly measured by droplet digital PCR (ddPCR) on single hepatocytes (Fig. 2b). Since mouse liver cells are long-lived (95% of hepatocytes remain quiescent over 1.5 years[25]), we assumed that cells do not divide throughout the simulation, but that genomes turnover at a consistent rate (mtDNA half-life is 9.4 days[26] giving 18.8 days for full replacement—a "Generation") with mutations continuing to occur during replication. Mutation emergence rate and selection coefficient ($s$, with $s$ = 0 for neutral, $s$ > 0 for positive selection, and $s$ < 0 for negative selection) varied in the simulations (see below and Methods). A mutation rate of $2 \times 10^{-8}$ per bp per replication cycle[27] and $s$ = 0 were used as default settings. Note, that since this model does not include cell division, events such as germline mutations and clones arising in early development are not reflected by the model. However, because maternal transmission and early emergence of specific alleles are rare events, they appear only

sporadically, allowing us to identify these events as unique occurrences in individual mice.

Figure 2c follows the simulated emergence and accumulation of a neutral mutant allele in cells over many cycles of mtDNA turnover (Generations). When a mutation first emerges, it is present as a single copy per cell amid 10,000 copies of mtDNA in a mouse liver cell. The abundance of newly emerged neutral mutations fluctuates over time, with the only stable outcomes being loss or fixation (all of mtDNAs in a cell contain a mutation)[28]. Loss was the predominant fate even after an initial rise (e.g., arrowhead in Fig. 2c), while fixation was seldom seen in our simulations. The rare mutations that rise to high abundance do so only after many cycles of mtDNA turnover (Fig. 2c). By varying parameters of our simulations, we examined how time (number of Generations), mtDNA copy number and mutation rate impact the likelihood that genetic drift would result in high abundance of a mutation (Supplementary Fig. 4). Importantly, the most influential parameters, time and copy number are determined from data. Figure 2d compares the expected distribution of abundances for neutral somatic mutations in hepatocytes of 24-month-old mice (red) to the actual data (blue). The observed abundances greatly exceed those predicted by the neutral model. We conclude that random drift of mutations emerging at rate of $10^{-9}–10^{-4}$ per bp per replication cycle cannot explain the accumulation of diverse mutations at high abundance. Having eliminated clones, we reason that predominance of abundant alleles could be attributed to an exceptionally high mutation rate, positive selection and/or some other previously unrecognized process.

## Positive selection

Single-cell analysis allows us to distinguish the influences of positive selection and mutation rate. Selection largely impacts the abundance of a mutation in the cells in which it occurs, while the mutation rate largely influences the number of cells in which a mutation occurs. To visualize these parameters, we determined the **a**verage cellular **a**bundance of each **a**llele (AAA) and plotted this on the *Y*-axis vs the number of cells (C#) in which the allele was detected on the *X*-axis (Fig. 3a, Supplementary Fig. 5a, d, Supplementary Fig. 6a and Supplementary Data 1). In this AAA vs C# plot, alleles with increasing mutation rates fall on a rising curve to the right (grey line in Fig. 3a and Supplementary Fig. 5a, and Supplementary Fig. 5b). In contrast, positively selected alleles rise to high cellular abundance and cluster toward the top of the graph (Supplementary Fig. 5c). In 24-month-old mice, indel alleles in homopolymers followed the grey line consistent with the known high rate of indels in such sequences. Alleles in the longest homopolymeric sequence in mouse mtDNA located in the OriL appeared to have the highest mutation rate (purple datapoints at the top of the grey curve in Fig. 3a, Supplementary Fig. 5a, d and Supplementary Fig. 6a). Characterization of these high-frequency indels will be described separately (in preparation). In contrast, a group of alleles in the non-coding region (NCR) clustered near the top of the plot and to the left suggesting that these alleles are positively selected (cyan datapoints in Fig. 3a, Supplementary Fig. 5a, d and Supplementary Fig. 6a).

Many of the NCR alleles located at the top of the AAA vs C# plot were detected repeatedly at high abundance in multiple cells and multiple aged, but not young, mice (Fig. 3a–c and Supplementary Fig. 6a–c), a feature distinguishing them from clonally amplified mutations which are sporadically detected at high abundance and localize above the grey line in AAA vs C# plot (e.g., synonymous mutation in Supplementary Fig. 5a, d). Repeated detection is expected not only for alleles with a high mutation rate, but also for alleles that are positively selected. According to neutral behavior, most newly emerged mutations are lost (Fig. 2c) and never rise to a level at which we can detect them. However, positive selection increases the likelihood that newly emerged alleles will evade elimination and, with time, amplify to detectable levels.

To further characterize the exceptional NCR alleles, we focused on those present at over 20% abundance in at least one cell in at least three out of five examined WT mice and in at least three cells in all three examined heterozygous mutator mice (Supplementary Fig. 7). These thresholds were set to exclude neutral and clonally amplified mutations. The thresholds for mutator mice were higher than those for WT due to their elevated mutation rate. We examined the abundance of each of these alleles in every cell in which it was detected (Fig. 3d) and matched the resulting abundance distributions to the array of simulated distributions produced by varying only mutation rate or in conjunction with different selection coefficients (Fig. 3e and Supplementary Fig. 8a). Making the simplifying assumption that mutation rates and selective forces are constants for each allele, this gave us approximate mutation rates and strength of selection for the NCR alleles in WT mice (Fig. 3e and Supplementary Fig. 8b). The estimated mutation rates ($10^{-8} – 3.16 \times 10^{-7}$) are close to previously reported rates determined in the human B cell line TK6 ($2 \times 10^{-8}$ per bp per cell division)[27], human colon ($5 \times 10^{-5}$ mutations per genome per day which translates into $2.5 \times 10^{-8}$ per bp per cell division assuming crypt stem cells divide once per three days)[29] and clones from human colorectal epithelium, hematopoietic stem cells and fibroblasts ($5 \times 10^{-8}$ per bp)[3]. Importantly, the experimental distributions of the exceptional NCR alleles could only be matched when a positive selection coefficient was introduced in the simulations.

Our simulations predict that neutral alleles would accumulate linearly in bulk tissue; in contrast, positively selected alleles are predicted to follow a power function (Fig. 3f, g). We developed ddPCR assays to measure the abundance of two NCR mutations (15468A>G and 16012G>A) in bulk tissue. Accumulation of these alleles in liver tissue taken from differently aged mice followed a power function (Fig. 3h, i), consistent with positive selection.

The positively selected NCR alleles are clustered in or close to DNA sequences important for mtDNA replication (Fig. 3j). As was previously argued from such associations in humans[15], we infer that the positive selection results from a replicative advantage incurred by the mutant genomes, although it should be clear that this advantage might be promoted in different ways including, but not limited to, altered efficiency of primer formation, pausing duration at the end of the NCR, or the successful termination and resolution of the replication products.

## Passenger mutations

Replicative advantage could explain the high abundance of a few alleles located in the NCR and extremely high mutation rate could account for the frequent indels located in the OriL. However, we observed many more alleles throughout the mtDNA at high abundance (Fig. 2d). Thus, we reasoned that there must be an additional process underlying the amplification of mutant mtDNAs.

Individual cells carrying positively selected NCR mutations occasionally harbored other mutations at similarly high levels (Fig. 4 and Supplementary Fig. 9). Such cases would be produced when, a positively selected NCR allele emerged on a genome with an existing mutation(s). The NCR allele could then act as a "driver" promoting the relative abundance of itself and of the linked "passenger" allele(s). In two of the three cells shown in Fig. 4, the NCR mutation and its passenger(s) were present at >50% of total mtDNAs, and therefore co-reside on at least some of the mtDNAs (Fig. 4a, c). In the third example, the physical linkage of one of several candidate passengers is clearly demonstrated by individual sequencing reads recording both the NCR mutation and the nearby passenger (Fig. 4g).

Since associations between a driver and any particular passenger are rare, examining the abundance of driver and passenger alleles in other sequenced cells should show their independent behavior (Fig. 4b, d, f). Driver alleles were found at abundant levels in multiple cells indicating an autonomous advantage. In contrast, when the

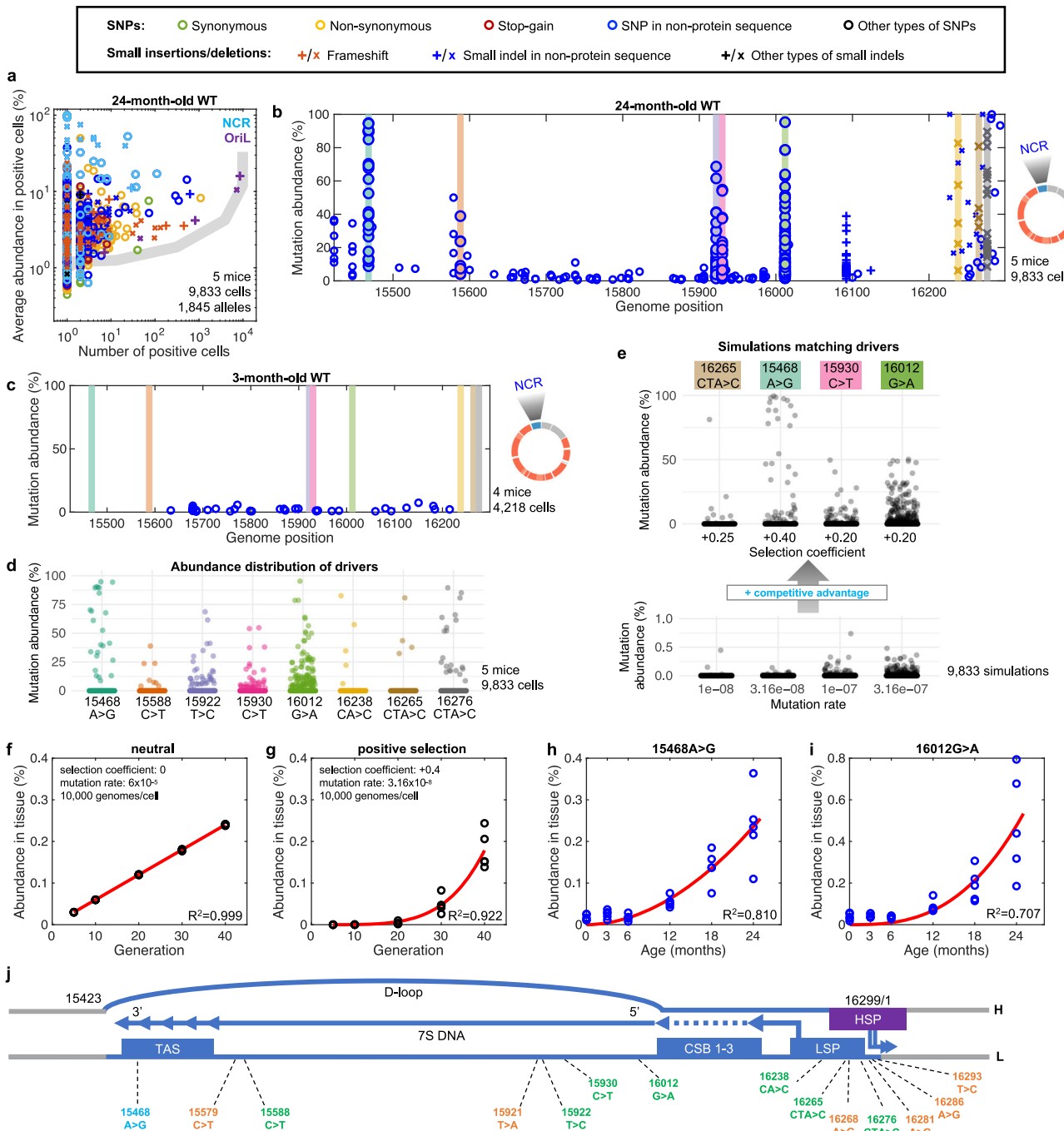

**Fig. 3 | Specific mutations in the NCR confer a competitive advantage. a** An AAA vs C# scatter plot for hepatocytes from 24-month-old WT mice. In this panel, in addition to above key, alleles in the NCR and OriL are colored cyan and purple, respectively. The grey line represents a simulation-based prediction for location of neutral alleles with varying mutation rates. **b** The spectrum of NCR mutations in hepatocytes from 24-month-old WT mice. Alleles present at over 20% abundance in at least one cell in at least three out of five mice are highlighted. **c** The exceptional NCR alleles (colored bars as in **b**) were not detected in four 3-month-old WT mice. **d** Abundance distribution of highlighted NCR alleles in cells of 24-month-old WT mice. **e** Simulations testing the influence of the parameters on the cellular distribution of mutation abundance. Adjusting the mutation rate (lower panel) alone cannot mimic the observed patterns (**d**), while the inclusion of positive selection coefficients improves the match (upper panel). Y-axis scales differ. **f**–**g** Simulations

show the linear accumulation of a neutral allele in whole tissues with time/Generations (**f**), while a positively selected mutant allele accumulates at an accelerating rate (**g**) **h**, **i** Measurement of two driver alleles using allele-specific ddPCR shows accelerating accumulation in bulk livers of WT mice. **f**–**i** N = 5 simulations or mice per time point. Red lines show linear (**f**) and power function (**g**–**i**) fitting. **j** Driver mutations localize in the replication initiation region: NCR (blue); the conserved sequence boxes (CSB1-3); the light strand (L) promoter (LSP), which primes DNA synthesis pausing at the termination-associated sequence (TAS); and the heavy strand (H) promoter (HSP). The allele labeled in cyan showed selective amplification in both WT and heterozygous mutator mice, whereas the alleles labeled in green and orange only passed the criteria for drivers in WT or heterozygous mutator mice, respectively. 16286A>G and 16293T>C alleles are likely drivers in WT mice that fell below our thresholds.

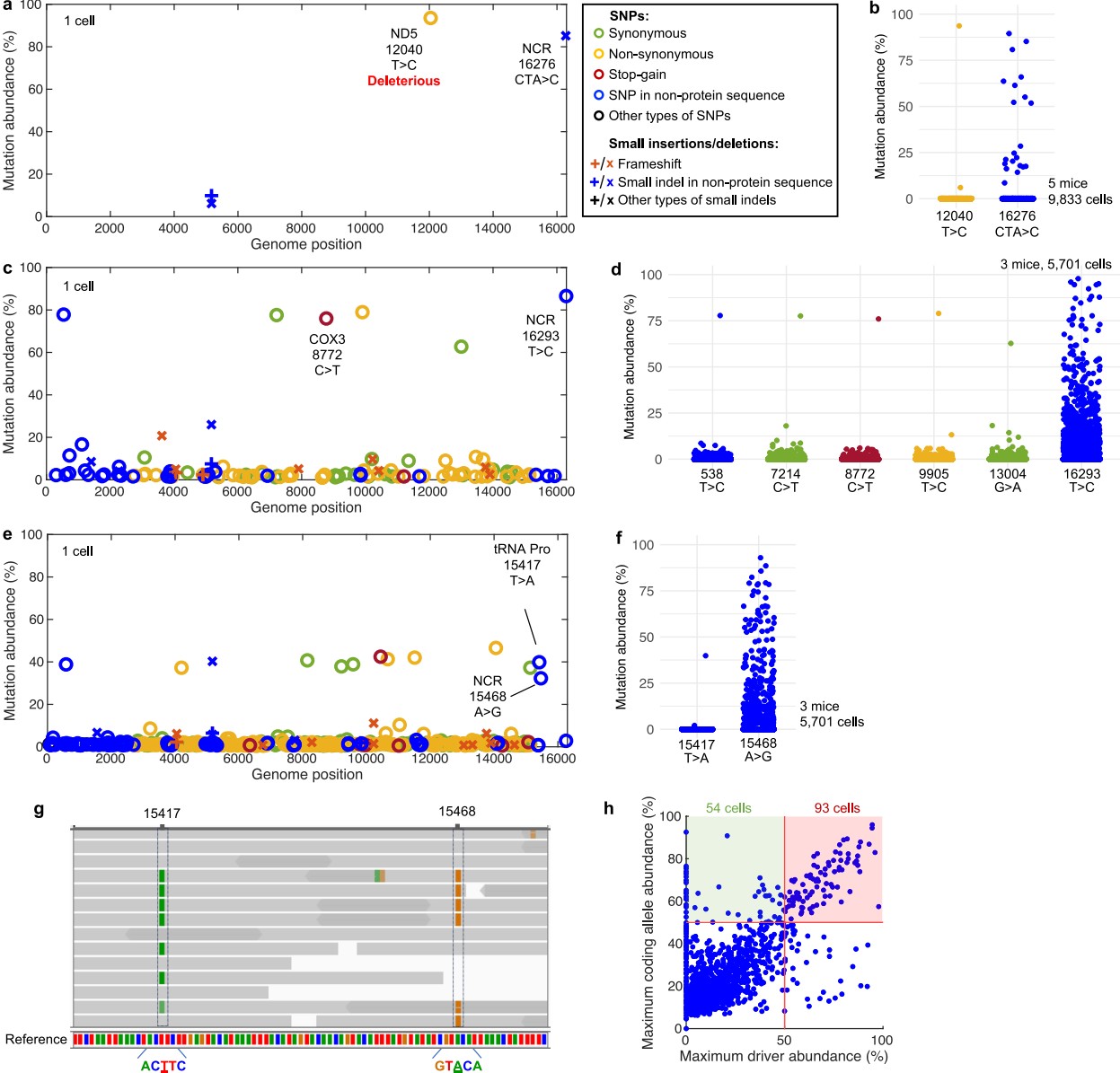

**Fig. 4 | Passenger mutations piggyback on mtDNAs with a competitive advantage. a** A mtDNA mutation spectrum of a single liver cell from a 24-month-old WT mouse showing two alleles at high abundance. **b** Abundance distribution of the 12040T>C and 16276CTA>C mutations among all sequenced cells. **c** A mtDNA mutation spectrum of a single liver cell from a 24-month-old heterozygous mutator mouse. **d** Abundance distribution of passenger and driver mutations shown in panel **c** among all sequenced cells from 24-month-old heterozygous mutator mice. **e** A mtDNA mutation spectrum of a single liver cell from a 24-month-old heterozygous mutator mouse. **f** Abundance distribution of the 15417T>A and 15468A>G mutations among all sequenced cells. **g** Raw reads showing the linkage of the 15417T>A and 15468A>G mutations. Grey lines represent individual reads. Dark grey regions represent the overlap of two opposing reads of the same DNA fragment. Green and orange bars mark mismatch between the read and reference sequences. **h** Maximum abundance level of driver alleles versus the maximum abundance level of coding alleles in a cell. Data are shown for 5701 hepatocytes from three heterozygous mutator mice. One abundant clonal mutation was excluded. Red lines are drawn at 50% abundance. Data points in the red square represent the linkage between a driver and a coding allele.

passenger allele occurred on its own, it was found at low abundance. This is consistent with a conclusion that in cells presented in Fig. 4 the passenger alleles gained an advantage by linkage to a driver allele.

The abundance of non-driver alleles that reach high abundance due to linkage with a driver, will be correlated with the abundance of their respective drivers. Therefore, to assess the scale of the impact of the "driver-passenger" effect, we plotted the maximum abundance level of driver alleles versus the maximum abundance level of non-driver coding sequence alleles for every cell from the aged heterozygous mutator dataset (Fig. 4h). This plot shows a strong diagonal indicating that many of the non-driver mutations that reach high abundance, do so by association with drivers. Since we selected

conservative thresholds for the identification of driver mutations, additional drivers might explain many of the off-diagonal abundant mutations (Fig. 4h). Notably, passenger alleles include missense mutations predicted to be deleterious (see Methods) as well as truncating mutations (Fig. 4a, c). Thus, by boosting the abundance of associated detrimental alleles, drivers are likely to compromise cellular metabolism.

## Selfish drive in the human liver

To test whether the selective forces we observed in mice operate in humans and to determine how they play out on a longer time scale, we profiled mtDNA mutations in human hepatocytes from six de-

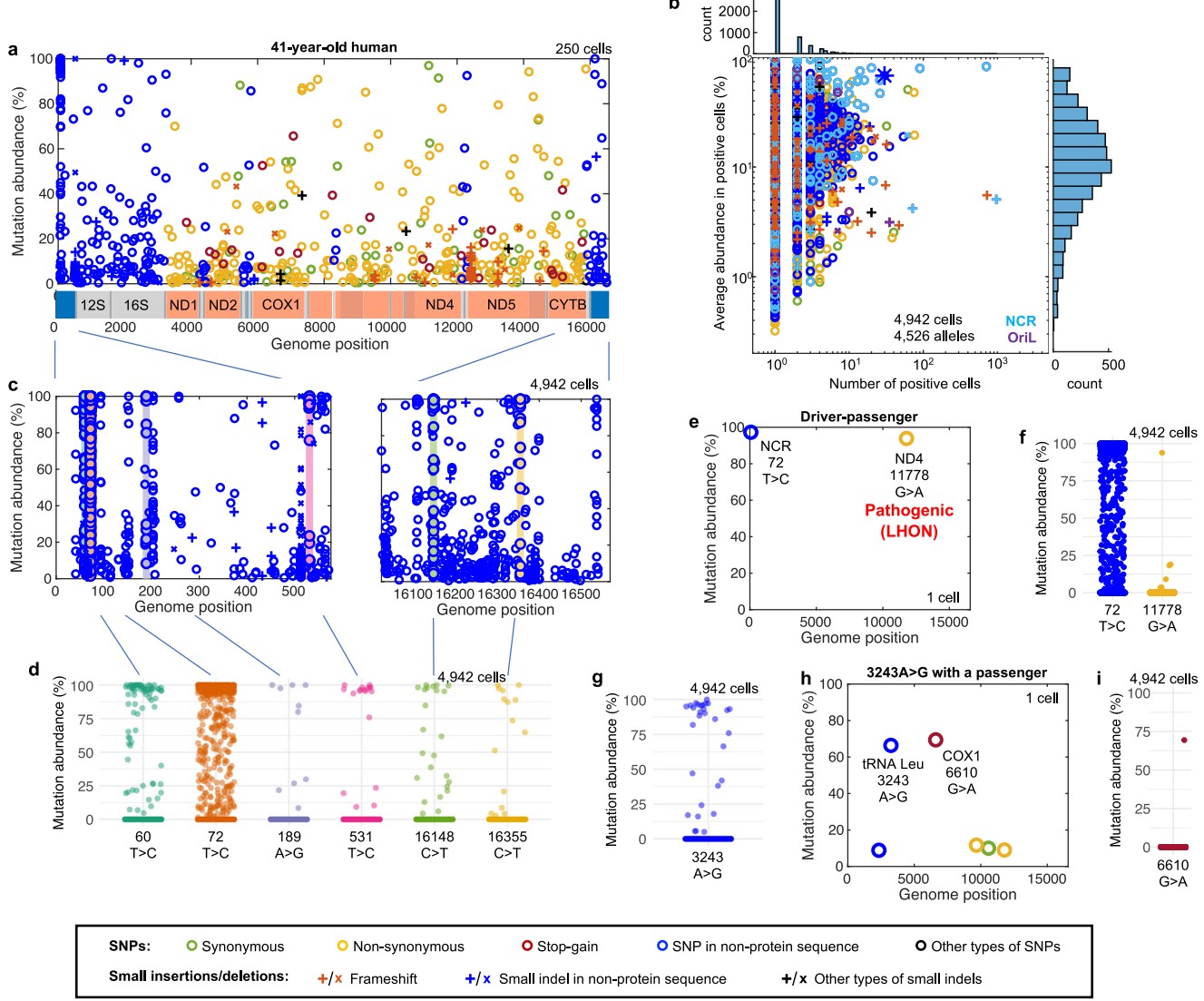

**Fig. 5 | Selective forces impact competition among mtDNAs in the human liver. a** A spectrum of mtDNA mutations identified in hepatocytes from a 41-year-old human. A subset of 250 cells is shown. Note that annotation of mouse and human mtDNAs differs, linearization of the human genome splits the NCR in two. **b** AAA vs C# plot of hepatocytes from the 41-year-old human. The blue asterisk marks the 3243A>G allele. **c** Magnified and annotated view of the NCR region using the complete dataset for the 41-year-old human showing the alleles (colored bars) classified as drivers in this individual (detected in ≥10 cells at ≥50% abundance with more cells exceeding 50% abundance than those below it). **d** Abundance distribution of the driver mutations identified in the NCR of hepatocytes from the 41-year-old human. **e** An example of a "driver-passenger" pair in a single liver cell from the 41-year-old human. **f** Abundance distribution of the driver and the passenger alleles shown in **e** among all 4942 sequenced cells of the sample. Note that data for the 72T>C allele are also a shown in **d**. **g** Abundance distribution of the 3243A>G mutation in liver cells of the 41-year-old human. **h** An example of a passenger mutation on 3243A>G genome. **i** Abundance distribution of the 6610G>A mutation in liver cells of the 41-year-old human.

identified individuals of known ages (Fig. 5a, Supplementary Fig. 10a, and Supplementary Data 2). As expected for a long-lived organism (Fig. 2c and Supplementary Fig. 4a), many more mutations accumulated to higher levels in aged human samples than in mice (Figs. 1c, 5a and Supplementary Fig. 10a). This is especially true for the oldest (81-year-old) human sample (Supplementary Fig. 10a, b) in which many mutations have abundances near 100% (most of these are likely fixed and fall short of 100% abundance due to measurement inaccuracies; Supplementary Fig. 2 and Methods). Two time-dependent mechanisms are expected to contribute to the high abundance of alleles: random drift and positive selection (Supplementary Fig. 10c, d).

As was the case with mouse data (Fig. 3a and Supplementary Fig. 5a), plotting the data in the abundance versus cell number (AAA vs C#) plot shows an enrichment of NCR alleles at the top of the graph (Fig. 5b, Supplementary Figs. 10e and 11). Moreover, the distribution

of cellular abundance of these alleles in individual cells matches that expected for driver alleles (Fig. 5c, d). Our data showed individual-to-individual variation in alleles exhibiting driver behavior (Supplementary Data 2). In total, we identified with high confidence 18 driver alleles within the NCR among the six samples analyzed. Several of these were previously identified as highly abundant alleles that occurred in a tissue-specific pattern in multiple individual people (Supplementary Data 2)[15,16,30,31]. We conclude that a selfish/replicative advantage contributes to the positive selection of NCR mutations in the human liver as we saw in mice. Finally, examination of mutations in single cells shows evidence of "driver-passenger" linkage as we described in mice (Fig. 5e, f).

Mutations that drive a replicative advantage are expected to climb to unusual abundance in the human population and, if detrimental, they could make a major contribution to mitochondrial disease. A well-

known SNP, 3243A>G, is the most prevalent genetic cause of mito-chondrial disease[32]. We tested the possibility that it might exhibit posi-tive selection. Unlike most studies investigating inherited 3243A>G mutation in patients, we examined the behavior of this mutation as it emerges in somatic tissues de novo using our single-cell DNA sequen-cing methodology. In total, we analyzed 25,997 hepatocytes from 6 individuals and detected the 3243A>G allele in 72 cells (-0.3%). Samples from all examined individuals included cells that displayed an unusually high abundance of the allele (Supplementary Fig. 11). In one individual, 3243A>G met our criteria for a driver allele despite being located out-side of the NCR (Fig. 5b, g and Supplementary Data 2). Furthermore, the distributions of 3243A>G abundance in the individual cells in which it was detected those produced by the inclusion of a positive selection coefficient (Fig. 5g and Supplementary Fig. 10d). Moreover, we identified instances when 3243A>G carried linked passenger mutations to high cellular abundance (Fig. 5h, i and Supplementary Fig. 12). These data together with previous results[33,34] suggest that 3243A>G is a special allele that possesses an intrinsic positive selection while also having a detrimental effect.

## Discussion

Our work reveals the unexpected impact of seemingly harmless mutations that boost the replication of affected genomes, indirectly leading to the significant accumulation of deleterious mutations. Below, we discuss the importance of selective processes acting on mtDNAs within cells in driving the emergence of mutant phenotypes and disease.

### A dangerous liaison

A cell can tolerate many recessive mtDNA mutations at low abundance since other genomes can provide WT proteins and RNAs. However, as cellular levels of mutant mtDNA rise, the remaining WT genomes cannot fully support the mitochondrial function, and phenotypes emerge. Thus, processes promoting a rise in the cellular abundance of mtDNA mutations render these mutations impactful. Our data-based modeling shows that, in cells with high mtDNA copy numbers and random mtDNA turnover, newly emerged mtDNA mutations are pre-dicted to remain at inconsequentially low levels even in aged indivi-duals. However, in our single-cell data, we observed multiple mutant alleles at high levels that could not be explained by the genetic drift. Some of these alleles rose to a high level due to an intrinsic positive selection, while others were hitchhiker mutations that would other-wise not accumulate to high levels within a cell.

Genetic studies in model organisms identified mtDNA variants with a propagation advantage. Such variants have been called selfish because they increase in abundance regardless of their impact on mitochondrial function[12]. While selfish variants can occur without causing any phenotypes, they were initially identified in fungal systems because they were associated with deleterious effects[10,11]. Studies of adult human tissues have also identified several relatively abundant mtDNA alleles proposed to have a replicative advantage[15]. However, these human mutations were not shown to have functional con-sequences, nor was their association with secondary passenger alleles considered.

The frequency at which high-abundance alleles are found to be associated with driver mutations in the mutator mouse (Fig. 4h) sug-gests that the association provides a major route promoting high abundance of mtDNA mutations. We propose the generalization that any positively selected allele can act as a driver to promote the rise of an associated mutation. We can think of the "driver-passenger" rela-tionship as a genetic catalyst promoting phenotypic variation by empowering otherwise silent mutations carried at inconsequential levels. The resulting enhancement of cellular phenotypes will promote the progressive worsening of symptoms in patients with mitochondrial disease and enhance the deterioration of mtDNA with age.

Driver-passenger associations will be especially impactful in individuals carrying a maternally inherited disease allele. In these individuals, a large number of cells throughout the body will harbor this specific allele. The presence of this allele at substantial levels in many cells increases the likelihood of driver-passenger associations amplifying the disease allele to detrimental levels. Indeed, among the few passengers we identified in aged WT mice, several were clonal mutations (Supplementary Fig. 9).

While our findings show that the "driver-passenger" pathway plays a major role in the increase of abundance of mtDNA mutations, there are likely to be other mechanisms of selection that also contribute. Additionally, because the impact of random genetic drift is higher in rapidly turning over cells with few copies of mtDNA, it will be impor-tant to understand the interaction of neutral variation with selective forces in various circumstances.

### Coevolution with the nuclear genome

Samuels et al.[15,31] emphasized the tissue specificity of the recurrent mutations they identified in the NCR of human mtDNA. A similar tissue-specificity of competition between two mtDNAs in a hetero-plasmic mouse led Jenuth et al.[35] to argue for "the existence of unknown, tissue-specific nuclear genes important in the interaction between the nuclear and mitochondrial genomes". If tissue-specific nuclear gene expression alters competition between mtDNAs, so might differences in the nuclear genome between individuals, a possibility supported by direct genetic identification of nuclear modifiers of mtDNA competition[36,37], and a genome-wide-association study in humans revealing nuclear loci associated with high abun-dance of particular mitochondrial alleles[38]. Indeed, human drivers vary from individual to individual as expected for an influence of genetic background[15,16,30] (Supplementary Data 2). For example, Samuels et al.[15] found the 16093T>C allele at high levels in multiple tissues in only one of two individuals, and we saw this allele as a strong driver mutation in the hepatocytes of one individual but not in the hepatocytes of five others (Supplementary Data 2). Addition-ally, while we found the same driver alleles repeatedly in different individuals within inbred strains of mice (C57BL/6J and mutator), among the sets of drivers we identified in the two strains, only one driver was shared (Fig. 3j).

Nuclear genes that promote replication of one mitochondrial genotype would disfavor other genotypes. Thus, a genome that is positively selected in one genetic background, can be negatively selected in another[36]. The extraordinary distinctions in the abundance of 16093T>C in different individuals are likely to include negative selection (i.e., a replicative disadvantage) in individuals in which the allele was undetectable (Supplementary Data 2). Finally, differences in nuclear gene expression associated with developmental stage, age, stress and diet are likely to alter the strength of selection for or against mtDNA loci sensitive to nuclear modification. Thus, while we have identified 18 driver alleles in the NCR in human hepatocytes from six individuals (Supplementary Data 2), a survey of additional tissues, individuals, and various life conditions is predicted to uncover many more.

It is reasonable to assume that the NCR sequence controlling replication has been optimized over evolutionary time scales, in which case one would not expect to find driver alleles. However, evolution would select for the optimal replicator only in the germ-line, and tissue-specific selection can "re-optimize" the NCR for replication in different somatic tissues. Since "re-optimization" of the NCR is the result of alterations of nuclear gene-action, changes in the nuclear genome or changes in nuclear gene expression will give new driver alleles a selective advantage. Outbreeding, which changes the genetic background, will widely trigger re-optimization of mtDNA for replication, including in the germline. Germline re-optimization, by triggering selection for new drivers, would promote germline

trapping and amplification of passenger alleles, possibly including disease alleles, that would be heritably transmitted.

### A disease allele with selfish drive

There are many mtDNA-associated disease alleles, but one, 3243A>G, is responsible for more cases of mitochondrial disease than any other identified mtDNA single-nucleotide change[32]. The 3243A>G mutation disrupts the gene *MT-TL1*, which encodes tRNA-leu (UUR)[33], as well as altering a sequence required for binding of a transcriptional terminator[39]. The mutation reduces mitochondrial translation[34], which is thought to be responsible for the deleterious consequences of the allele. However, other mutant alleles share these molecular defects yet lack the high incidence of 3243A>G mutation, leaving us without an explanation for its disproportionate prevalence.

Our study examined the behavior of this mutation as it emerges in somatic tissues de novo. Despite an origin from a somatic mutation, on average, 3243A>G reached 50% abundance in the rare positive cells, which translates into a staggering ~2500-fold increase in abundance from the initiating event. Our data indicate that 3243A>G is positively selected, which is consistent with a previous observation[40], but it is not clear by what mechanism it gains a replicative advantage.

Apparently contradicting our finding, in patient blood, 3243A>G levels are especially low and decline with age, suggesting that 3243A>G is under negative selection[41–43]. This negative selection appears to be the consequence of selection at the cell level in T-cells[44,45]. 3243A>G tissue abundance was also shown to decline in other mitotic tissues but remain high in post-mitotic tissues[46,47]. We propose that positive intracellular selection of 3243A>G acts in at least some cell types and that it is countered by negative cellular selection, which is most powerful in mitotic tissues. Importantly, positive selection of 3243A>G in the germline in at least some genetic backgrounds[48] could account for its prevalence in the human population. Modification of tissue-specific selection by genetic background could contribute to its diversity of disease presentations.

## Methods

### Animals
C57BL/6J, mtPWD (C57BL/6J-mt^PWD/Ph^/ForeJ) and mutator (B6.129S7(Cg)-Polg^tmlProl^/J) mice were obtained from The Jackson Laboratory. Three to 24-month-old male and female mice were used. To avoid a contribution of maternal mtDNA mutations, heterozygous mutator males were bred with C57BL/6J females to produce heterozygous progeny for experiments. mtPWD mice were maintained by crossing mtPWD females with C57BL/6J males. Mice were housed in a specific pathogen-free facility with a standard 12-h light/dark cycle at the University of California, San Francisco, and given food and water *ad libitum*. Temperature in the facility was maintained between 67 °F and 74 °F, humidity was maintained between 30 and 70%. Experiments were conducted in accordance with institutional guidelines approved by the University of California, San Francisco Institutional Animal Care and Use Committee, approval number: AN197290-01.

### Human hepatocytes
Cryopreserved deidentified human hepatocytes were purchased from Xenotech, Lonza or UCSF Liver Center. Samples were deidentified by the provider and therefore work with such samples is not considered human subject research. The limited available clinical information about analyzed human individuals is reported in Supplementary Data 2. Work with living human cells was conducted at biosafety level 2.

### ddPCR
Genomic DNA was isolated from 25 mg of liver tissue using DNeasy Blood and Tissue kit (Qiagen, 69506) according to manufacturer's guidelines. Primers and probes were synthesized by Integrated DNA Technologies, Inc. (IDT) and their sequences are provided in Supplementary Data 3. WT C57BL/6J mtDNA sequence (positions 15,196-136) was cloned into pGEM-T (Promega, A1360) vector and used as WT control. A 500 bp fragment of mouse mtDNA containing 15468A>G or 16012G>A mutations was synthesized by IDT and cloned in pUCIDT-AMP vector to use as positive control. The ddPCR reaction mixture contained ddPCR Super Mix for Probes (Bio-Rad, 1863024), 900 nM of forward primer, 900 nM of reverse primer, 250 nM of WT probe, 250 nM of mutant probe, 0.5 μL of restriction enzyme (HaeIII; NEB, R0108L) and template DNA. Template DNA concentration was adjusted to be below 3500 mtDNA copies per microliter of ddPCR reaction mixture. 20 μL of the reaction mixture and 70 μL of oil (Bio-Rad, 1863005) were loaded on a DG8 cartridge (Bio-Rad, 1864007) for droplet generation on QX100 Droplet Generator (Bio-Rad). 40 μL of droplet emulsion were transferred to 96-well plate (Bio-Rad, 12001925) and sealed with a pierceable foil (Bio-Rad, 1814040) using PX1 PCR plate sealer (Bio-Rad). The optimized PCR thermal cycling was conducted on a conventional PCR machine (Bio-Rad, C1000 Touch). Thermocycling conditions for the 15468A>G assay: 10 min polymerase activation at 95 °C; 40 cycles of denaturation at 94 °C for 30 s, ramp rate 1 °C/s, and combined annealing-extension at 54 °C for 2 min, ramp rate 1 °C/s; incubation at 98 °C for 10 min. Thermocycling condition for the 16012G>A assay: 10 min polymerase activation at 95 °C; 45 cycles of denaturation at 94 °C for 30 s, ramp rate 1 °C/s, and combined annealing-extension at 52 °C for 2 min, ramp rate 1 °C/s; incubation at 98 °C for 10 min. After thermocycling, samples were cooled to room temperature and analyzed on the QX100/200 Droplet Reader (Bio-Rad). Results were analyzed with QuantaSoft Analysis Pro v.1.0.596 software (Bio-Rad).

### Hepatocytes isolation
Mouse hepatocytes were isolated by a two-step perfusion technique. Briefly, mouse was anesthetized by isoflurane (Piramal Critical Care). Mouse liver and heart were exposed by opening the abdomen and cutting the diaphragm away. The portal vein was cut, and immediately the *inferior vena cava* was cannulated via the right atrium with a 22-gauge catheter (Exel International, 26746). Liver was perfused with liver perfusion medium (Gibco, 17701-038) for 3 min and then with liver digest medium (Gibco, 17703-034) for 7 min using a peristaltic pump (Gilson, Minipuls 3). Pump was set to 4.4 mL/min and solutions were kept at 37 °C. After perfusion the liver was dissected out, placed in a petri dish with hepatocyte plating medium (DME H21 [high glucose, UCSF Cell Culture Facility, CCFAA005-066R02] supplemented with 1x PenStrep solution [UCSF Cell Culture Facility, CCFGK004-066M02], 1x Insulin-Transferrin-Selenium solution [Gibco, 41400-045] and 5% Fetal Bovine Serum [UCSF Cell Culture Facility, CCFAP002-061J02]) and cut into small pieces. Liver fragments were passed through a sterile piece of gauze. Hepatocytes were separated from non-parenchymal cells by centrifugation through 50% isotonic Percoll (Cytiva, 17-0891-01) solution in HAMS/DMEM (1 packet Hams F12 [Gibco, 21700-075], 1 packet DMEM [Gibco, 12800-017], 4.875 g sodium bicarbonate, 20 mL of a 1 M HEPES pH 7.4, 20 mL of a 100X Pen/Strep solution, 2 L H$_2$O) at 169 g for 15 min. Isolated hepatocytes were used immediately for FACS or frozen in BAMBANKER (GC LYM-PHOTEC, CS-02-001) and stored at -80 °C for future experiments.

### Cell sorting
Isolated hepatocytes were resuspended in PBS (Gibco, 10010-023), stained with 5 μg/mL propidium iodide (Invitrogen, P1304MP) to mark dead cells, and kept on ice until FACS. Right before sorting hepatocytes were strained through 35–40 μm cell strainer. Sorting was performed on FACSAriaII (Becton Dickinson) using 100 μm nozzle. Instrument was calibrated using 23.9 μm beads (Spherotech, ACURFP2.5-250-5). Single hepatocytes were sorted into 384-well plates

(Bio-Rad, HSP3801 or 4titude, 4ti-0384) containing 0.45 µL of TD buffer (10 mM TrisHCl pH 8.0, 5 mM MgCl₂, 10% dimethylformamide [Acros Organics, 423640250]). Due to their large size and extreme size variability, sorting of mouse hepatocytes was inefficient and only 40–60% of wells contained cells while the rest of the wells were empty. Human hepatocytes sorting efficiency was ~90%. One column of a plate was left empty to serve as a negative control. Immediately after sorting plates with hepatocytes were sealed with foil (Bio-Rad, MSF1001 or 4titude, 4ti-0500FL), briefly centrifuged, frozen on dry ice and stored at −80 °C.

## Single cell ddPCR
mtDNA copy number was quantified using single cell ddPCR. Frozen 384-well plate with single hepatocytes in 0.45 µL of TD buffer was thawed on ice. To lyse hepatocytes 0.45 µL of solution containing 10 mM Tris-HCl pH 8.0, 50 mM NaCl, 40 ng/µL MS2 RNA, 0.4% SDS and proteinase K (8 U/ml; NEB, P8107S) was added to 96 wells of the plate with help of acoustic liquid handler Echo 525 (Beckman Coulter). Wells without cells were used to prepare positive and negative controls. gBlock encompassing amplified sequence was synthesized by IDT and used as a positive control. After lysis and control solutions were added, the plate was sealed, briefly centrifuged, and incubated at 50 °C for 15 min and then at 95 °C for 10 min. After lysis, ddPCR master mix (ddPCR Super Mix for Probes (Bio-Rad, 1863024), 250 nM of forward primer, 250 nM of reverse primer, 250 nM of probe, 0.5 µL of restriction enzyme [AluI (NEB, R0137L) for mouse assay and HaeIII (NEB, R0108L) for human assay]) was added to each of 96 wells (final volume 22 µl), plate was sealed and vigorously vortexed, briefly centrifuged and incubated at 37 °C for 15 min to digest DNA. After restriction enzyme digestion, the plate was vigorously vortexed, briefly centrifuged and 20 µL of the reaction mixture was used for ddPCR as described above. Primers and probes sequences are provided in Supplementary Data 3. Thermocycling conditions for mouse mtDNA copy number assay: 10 min polymerase activation at 95 °C; 40 cycles of denaturation at 94 °C for 30 s, ramp rate 2 °C/s, and combined annealing-extension at 52 °C for 1 min, ramp rate 2 °C/s; incubation at 98 °C for 10 min. Thermocycling conditions for human mtDNA copy number assay: 10 min polymerase activation at 95 °C; 40 cycles of denaturation at 94 °C for 30 s, ramp rate 2 °C/s, and combined annealing-extension at 56 °C for 1 min, ramp rate 2 °C/s; incubation at 98 °C for 10 min. Due to inefficient FACS sorting some of the wells were empty, wells with fewer than 100 mtDNA copies were considered negative and excluded from the analysis.

## Plate-based single cell mtATAC
Frozen 384-well plates with single hepatocytes were thawed on ice. Hepatocytes were lysed and DNA was tagmented in a single step. To this end, 0.45 µL of lysis solution (1% n-Dodecyl β-D-maltoside [Sigma, D5172; final concentration 0.5%], 90 mM NaCl [final concentration 45 mM], 10 mM TrisHCl pH 8.0, 5 mM MgCl₂, 10% dimethylformamide) supplemented with Tn5 (Illumina, 20034197; 1.5 µL of enzyme for 150 µL of lysis solution) was added to each well of a plate with help of Echo 525. Then, plates were sealed with foil (Bio-Rad, MSB1001), briefly centrifuged, and incubated at 37 °C for 30 min. After lysis and tagmentation, Tn5 was stripped off DNA. To this end, 0.1 µL of 2% SDS was added to each well of a plate (final concentration 0.2%) using Echo 525, plates were sealed with a foil, briefly centrifuged, and incubated at 65 °C for 15 min. Next, mtATAC libraries were constructed by PCR amplification of DNA fragments created by Tn5 with unique dual index primers for each well of a plate. To this end, PCR master mix (NEB, M0544S), tween-20 (to quench SDS; final concentration 0.34%) and unique dual index primers (final concentration 500 nM; sequences are provided in Supplementary Data 3; IDT) were added to each well of the plate using Echo 525 (final volume 3 µL), plate was sealed with a foil, briefly centrifuged and thermocycled as follows: incubation at 72 °C

for 5 min to fill the gaps; initial denaturation at 98 °C for 30 s; 16 cycles of denaturation at 98 °C 10 sec and combined annealing-extension at 65 °C 75 s; final extension at 65 °C 5 min. Incubation and PCR were performed in a standard thermocycler (Bio-Rad, C1000 Touch or S1000). Uniquely labeled libraries from one or several plates were pooled together at equal volumes and cleaned up using home-made SPRI beads twice[49]. The first cleanup was one-sided with 1.2 beads to library volume ratio. The second cleanup was two-sided with 0.5 ratio followed by 1.2 ratio. Cleaned up libraries were eluted in 20 µL of TE buffer. To quality control and quantify libraries 1 µL of cleaned mtATAC library was run on Bioanalyzer (Agilent). mtATAC libraries were sequenced on MiSeq (Illumina) using MiSeq Reagent Kit v2, 300-cycles (Illumina, MS-102-2002) as 151 × 12 × 12 x 151.

This method was used to generate data presented in Fig. 1c, f and Supplementary Figs. 1, 3a–d.

## 10X-based single cell mtATAC
Frozen hepatocytes were thawed, washed with PBS and fixed in 1% PFA (Electron Microscopy Sciences, 15714-S) for 10 min at RT. After fixation PFA was quenched with glycine (125 mM final concentration) and washed with cold PBS supplemented with 1% BSA (Sigma, A1933). Next, hepatocytes were permeabilized. To this end, 1 million fixed cells were resuspended in 200 µL of lysis solution (0.5% n-Dodecyl β-D-maltoside, 45 mM NaCl, 10 mM Tris-HCl pH 8.0, 5 mM MgCl₂, 10% dimethylformamide) and incubated on ice for 5 min. For human hepatocytes n-Dodecyl β-D-maltoside concentration was reduced to 0.1%. Permeabilization was stopped by adding 1.8 ml of wash buffer (45 mM NaCl, 10 mM Tris-HCl pH 8.0, 5 mM MgCl₂, 1% BSA). To enable pooling of distinct samples in a single 10X experiment, permeabilized cells from different mice were labeled with unique DNA barcode complexes (MULTI-ATAC[21]). MULTI-ATAC barcoding was also performed when cells from a single individual were analyzed to allow identification of multiples, which is critical for our study and virtually impossible to achieve with other methods such as AMULET (a computational approach that relies on number of fragments)[50] due to variable ploidy of hepatocytes. In this case, an individual sample was divided into three to seven fractions and each fraction was labeled with a unique MULTI-ATAC barcode. To this end, Lignoceric Anchor oligo (2 µM; Sigma, LMO001A) was mixed with a unique barcode oligo (1 µM; BC; Supplementary Data 3) and reverse primer (1 µM; BE; Supplementary Data 3) at 1:1:1 molar ratio to form Anchor-BC-BE Complex (20x, 1 µM). Note that BC contains an 8-nucleotide-long stretch of random nucleotides to serve as a unique molecular identifier (UMI) to enable barcode counting. Permeabilized hepatocytes were resuspended at 10⁶ cells/mL in cold PBS and Anchor-BC-BE Complex was added to cell suspension (final 1x, 50 nM) followed by incubation on ice for 5 min. To stabilize labeling, Palmitic Co-anchor oligo (2 µM, 20x; Sigma, LMO001B) was added to the cell suspension (final 1x, 50 nM) followed by additional incubation on ice for 5 min. After MULTI-ATAC barcoding, unbound complexes were washed away with PBS supplemented with 2% BSA, hepatocytes isolated from different individuals were pooled together, resuspended in diluted nuclei buffer (10X Genomics, 2000207), passed through 35–40 µm cell strainer, and used to prepare 10X-mtATAC libraries using Chromium Next GEM Single Cell ATAC Reagent kit (10X Genomics, PN-1000176 and PN-1000406) according to the manufactures protocol (CG000209 Rev F and CG000496 Rev B) with 2 minor modifications. First, after step 3.2o 1 µL of the sample was used to prepare the MULTI-ATAC barcode library (described below). Second, the remaining 39 µL were used in step 4.1 where SI-PCR Primer B concentration was increased to 100 µM. Before permeabilization, after permeabilization and after MULTI-ATAC barcoding cells were pelleted by centrifugation at 100 g for 3 min, 300 g for 3 min and 500 g for 5 min, respectively.

To prepare MULTI-ATAC barcode libraries, 1 µL of sample from 3.2o step was amplified in a PCR reaction: 1 µL of sample, 500 nM SI-

PCR-B primer, 500 nM TruSeq primer (IDT, Supplementary Data 3), 1x Kapa HiFi HotStart ReadyMix (Roche, KK2601). The reaction mixture was thermocycled using the following conditions: 5 min polymerase activation at 95 °C; 14 cycles of denaturation at 98 °C for 20 s, annealing at 67 °C for 30 sec and extension at 72 °C for 20 s; incubation at 72 °C for 1 min.

To quality control and quantify libraries, 1 µL of 1:5 diluted 10X-mtATAC and MULTI-ATAC barcode libraries were run on Bioanalyzer. 10X-mtATAC and MULTI-ATAC barcode libraries were pooled together and sequenced on NovaSeq6000, S1 200 as 101x12x24x101 or NovaSeq X, 10B as 51x12x24x51 or 151x12x24x151. For optimal demultiplexing, we aimed to obtain 5000 MULTI-ATAC barcode reads per cell.

This method is prone to low-level leakage of mutation signal between cells (Supplementary Fig. 2). Since inbred mice have identical mtDNAs and mutations are very rare, this leakage becomes noticeable only if clonal mutations are present. Unlike inbred mice, humans have multiple haplotype- and individual-specific mtDNA variants. Consequently, leakage is noticeable at multiple sites if different human samples are mixed in a single experiment. Therefore, to simplify downstream analysis, all human samples were processed individually. Importantly, leakage also affects our readings of fixed mutations: as the predominant WT signal leaks into cells with fixed mutations, often we detect these mutations at levels just below 100%.

This method was used to generate data presented in Figs. 1d, e, 2d, 3a–d, 4, 5 and Supplementary Figs. 2, 3e, 5a, d, 6, 7, 9, 10a, b, e, 11, 12.

### Sequencing data analysis
**Reads mapping, coverage, and variant analysis.** The nucleus contains multiple segments derived from mtDNA sequence, so called NUMTs. When sequencing reads from ATAC experiments are aligned to the whole genome a lot of truly mitochondrial reads are erroneously mapped to the NUMTs. To avoid incorrect mapping of mitochondrial reads to the nuclear genome sequencing reads from the plate-based approach were aligned directly to the mtDNA. Specifically, reads were aligned to the mouse mtDNA (NC_005089) with *bwa*[51] (v0.7.17) using the *BWA-MEM* algorithm. Samples with less than 10,000 reads mapping to mtDNA (chrM) were excluded from further analysis. Duplicate reads were marked with *Picard tool*[52] (v2.27.4). Mapped reads were filtered with *bamtools*[53] (v2.5.2; -mapQuality ">= 20" -isPaired "true" -isProperPair "true"). Coverage was determined with *samtools depth*[54] (v1.16.1). SNPs and small indels were called using *Freebayes*[55] (v1.3.6; -C 5 -F 0.003 -p 1 --pooled-discrete --pooled-continuous -m 30 -q 30 --min-coverage 10). Multiallelic sites were split into multiple rows using *bcftools*[54] (v1.16; norm -Ov m-both). Variants were filtered using *vcffilter*[56] (vcflib v1.0.3; -f "SAF > 1" -f "SAR > 1"). Complex alleles were reduced to primitive alleles using *vcfallelicprimitives* and sorted with *vcfstreamsort*[56] (vcflib v1.0.3). This process occasionally created duplicate variants where mutation counts were split between the records which led to incorrect mutation frequency calculation. This issue was fixed by merging duplicated records in a single entry with alternative allele counts summed together. This was done after vfc files from individual cells were merged using *bcftools* (merge -m none) and the resulting vcf file was converted to tab delimited file using *vcf2tsv*[56] (vcflib v1.0.3). The variants were spot checked in IGV[57,58] (v 2.4.16). Variants annotation (synonymous, non-synonymous, stop-gain and etc.) was done using *SnpEff*[59] (v5.0). When the same variant had multiple annotations (e.g., due to overlap of protein coding sequences) the most severe annotation was used. *SIFT 4G*[60] was used to predict whether a mutation affects protein function.

Sequencing data from 10X-based experiments were first processed with *Cell Ranger ATAC* (10X, v 2.1.0) using blacklisted reference genomes[18]. Blacklisting was necessary to prevent erroneous mapping of mtDNA fragments to the nuclear DNA. Because our samples are non-standard and predominantly contain mtDNA reads, *Cell Ranger ATAC* does not discriminate well between empty droplets and droplets

containing cells. To classify droplets into those containing cells and those that are empty, as well as to assign cells to samples in multiplexed experiments and identify droplets with multiple cells, we relied on MULTI-ATAC barcode UMI counts. Barcode UMI counts have a bimodal distribution where positive (high-count) and negative (low-count) droplets for a specific barcode are clearly separated. A list of droplets that contained at least 1000 or more reads that passed filters (metrics provided by *Cell Ranger ATAC*; the cutoff was set to include 1.5–2 times more droplets than expected recovery) along with MULTI-ATAC barcode library FASTQ files were supplied to *deMULTIplex2*[61] (v1.0.1). For proper performance, *deMULTIplex2* requires removal of most empty droplets before demultiplexing. Hence, the barcodes count matrix created by deMULTIplex2 was filtered based on total number of MULTI-ATAC barcode UMIs. The cutoff was determined by plotting a histogram of barcodes counts and finding the middle between the two peaks representing positive and negative droplets. Whenever possible faithfulness of demultiplexing was controlled by analysis of distribution of sample-specific SNPs among multiplexed samples. Reads from droplets that were classified as carrying a single cell were subset from *possorted_bam.bam* file generated by *Cell Ranger ATAC*[54] into separate bam files using *samtools*[54] (v1.16.1). Reads deduplication, reads filtering, variant calling, variant filtering and variant annotation were the same as for plate-based approach. Cells with average mtDNA (chrM) coverage less than 50 were excluded from the analysis.

In addition to standard variant filtering, the following calls were excluded from the final datasets. Large-scale deletions in minor and major arcs of mouse mtDNA cause misalignment at the imperfect repeat regions creating false 4920C >T and 4925C>G, and 8686T>C, 14251T>A and 14260T>G mutations, respectively, that were excluded from the final dataset. Due to high number of mismatches between C57BL/6J mtDNA (NC_005089) and PWD mtDNA (DQ874614) sequences in NCR (positions 15,400-15,600), reads from mtPWD samples do not map to C57BL/6J reference in this region. Therefore, PWD-specific SNPs in this region were excluded from the dataset when analyzing C57BL/6J and mtPWD mixing results. Human mtDNA reference sequence (NC_012920) contains *N* at position 3107, which denotes a deletion. This *N* is misinterpreted by the aligner as any nucleotide which leads to 3107N>C, 3107N>T and 3109T>C false mutations. These calls were excluded from the final dataset. The region between positions 300 and 320 of human mtDNA had low coverage and multiple sequencing errors making it difficult to distinguish true and false mutations. Therefore, variants in this region were removed from the dataset. Finally, variants with mean abundance above 90% in a human sample were considered haplotype or individual-specific polymorphisms and were excluded from the list of mutations.

Despite our best effort to remove false calls from the dataset, there are some remaining artifacts, particularly errors in sequencing or alignment at the ends of the reads. These usually are present at very low levels and are unlikely to have any impact on our conclusions. All the specific mutations (such as driver and passenger alleles) that we rely on to draw conclusions were hand checked in IGV.

**Percent of reads mapping to mtDNA.** The total number of reads and number of reads mapping to mtDNA (chrM) were calculated using *samtools view*. Number of reads mapped to mtDNA was divided by total number of reads and the values were converted into %.

**Number of mutant alleles.** To calculate number of mutant alleles (Supplementary Fig. 3e) we used data produced with 10X-based approach. The number of unique mutant alleles strongly depends on coverage and number of analyzed cells. To mitigate biases due to coverage differences between samples we subsampled deduplicated and filtered BAM files for individual cells to 100,000 mtDNA (chrM) mapped reads. Cells that had fewer reads were excluded from the

analysis. Subsampled files were used for variant calling as described above. Finally, we normalized samples by analyzing equal number of cells from each sample.

## Modeling

To model accumulation of de novo somatic mtDNA mutations in hepatocytes we used an evolutionary simulation framework *SLiM*[24] (v3.6). We regarded mtDNAs as individuals and all mtDNAs within a single cell as a population. Since mouse hepatocytes are long-lived[25] we assumed that cells do not divide over the course of simulations. Modelling was done following authors recommendations[62] (Section 14.9). The following parameters were used:

1. mtDNA reproduces clonally.
2. The recombination rate was set to zero.
3. Population size was kept constant and set to the number specified in a figure or figure legend. Generally, population size of 10,000 genomes was used for modeling mouse hepatocytes and population size of 5000 genomes was used for modeling human hepatocytes.
4. The half-life for mtDNA in rat liver was estimated to be 9.4 days[26], which translates into 39 replacement cycles (Generations) over 2 years (maximum age of analyzed mice) and 1573 Generations over 81 years (maximum age of analyzed human samples). We assume that in mouse and human the half-life of mtDNA is similar to what was measured in rat and, for convenience, round it to 40 and 1600 Generations for 2-year-old mouse and 81-year-old human, respectively.
5. The mutation emergence rate and selection coefficient varied in the models and are specified in figures and/or figure legends. The mutation rate of $2 \times 10^{-8}$ per bp per replication cycle[27] and $s = 0$ were used as default settings. Note that this model does not include back-mutations. Hence, simulations using high mutation rates will be progressively less accurate.
6. To model the emergence, disappearance, and accumulation of a specific mutant allele, we simulated one site per genome and ran the simulation many times (Figs. 2c, 3e and Supplementary Figs. 5b, c, 8, 10c, d). To model accumulation of mutations in the whole genome, genome size was set to 16,299 bp (Fig. 2d and Supplementary Fig. 4). To simulate accumulation of a specific mutant allele at the tissue level we first simulated accumulation dynamics of the allele in 10,000 single cells and then computed the average abundance of the mutant allele across all simulated cells (Fig. 3f, g).

**Parameter space exploration.** To find parameters that best describe the behavior of recurrent NCR mutations in 24-month-old mice we searched the parameter space. Specifically, we ran 315 models where mutation rate varied from $10^{-9}$ to $10^{-2}$ and selection coefficient varied from −0.5 to +0.5. Each simulation was run for 40 generations, population size was set to 10,000 genomes and it was repeated 9833 times to match number of sequenced cells. For simulated data we have a record of all mutations present in a modeled cell, however in real sequencing data we lack mutations present at levels below detection threshold. To mimic the observed data, each of 9833 simulated cells was randomly assigned coverage of one of the cells from the experimental dataset. Then, simulated mutations present at levels below a sensitivity cutoff were set to zero. For each NCR mutation examined a sensitivity cutoff was calculated as $100\% \times 5/$(assigned coverage at this site), where 5 is a minimum number of reads supporting the mutation. The resulting abundance distributions from each of 315 models were compared to the observed distribution of a driver NCR mutation. First, the parameter sets that produced two times more or two times fewer positive cells than was observed were excluded. Next, we assessed the statistical significance of differences in abundance means between simulated and observed data using a permutation test. To calculate

mean we used only cells/simulations that were positive for a mutation. 1000 permutations were run to obtain a *p* value. Parameter sets with the highest *p* value produced the best data-matching distributions. Note that simulation outcomes in Fig. 3e are shown without adjustment for sensitivity.

### Statistical analysis and data visualization

Statistical analysis and data visualization were performed in Matlab (v. R2019a) and R (v. 4.1.3). Sample sizes, statistical tests and *p* values are indicated in the text, figures and figure legends.

### Reporting summary

Further information on research design is available in the Nature Portfolio Reporting Summary linked to this article.

## Data availability

The sequencing data generated in this study have been deposited in the NCBI Sequence Read Archive under PRJNA1146058. mtDNA copy numbers in single hepatocytes are provided in the Source Data file. Source data are provided with this paper.

## Code availability

No original code was used in this study.

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

## Acknowledgements

We are grateful to Saul Villeda for sharing aged mice and support with establishment of mutator mouse colony. We thank UCSF LARC for help with mouse husbandry. We thank Spyros Darmanis (CZ Biohub) for sharing index primer sequences for mtATAC library preparation. We thank Eric Chow and UCSF CAT for providing access to basic and cutting-edge equipment, support and advice, and Steven Deluca for the suggestion to use ATAC-seq for mtDNA profiling. Sequencing was performed at the UCSF CAT, supported by UCSF PBBR, RRP IMIA, and NIH 1S10OD028511-01 grants. This study was supported in part by the Liver Cell Isolation, Analysis & Immunology Core of the UCSF Liver Center (P30DK026743) and HDFCCC Laboratory for Cell Analysis Shared Resource Facility through a grant from NIH (P30CA082103). Portions of this work were performed on the Wynton HPC Co-Op cluster which is supported by UCSF research faculty and UCSF institutional funds. We thank the UCSF Wynton team for their ongoing technical support of the Wynton environment. This work was funded by Larry L. Hillblom Foundation (2018-A-028-FEL to E.K.; 2019-A-011-NET to P.H.O'F.; 2019 John S. Spice award in Aging to E.K.), UCSF Program for Breakthrough Biomedical Research (2019-2020 New Frontier Research Award to P.H.O'F. and Saul Villeda; 2021-2022 Postdoc Independent Research Grant to E.K.), NIH (R35GM136324 to P.H.O'F. and R33CA247744 to Z.J.G.) and CNV Stiftung (2019-2021 stipend to E.K.). We thank Chun-Yi Cho, Alexander-Sandy Johnson, Hansong Ma, Hiten Madhani and Barbara Panning for critical reading of the manuscript.

## Author contributions

E.K. and P.H.O.'F. conceived the project, designed experiments, interpreted the results, and secured funding. E.K. and D.N.C. adapted MULTI-ATAC and 10X scATAC for profiling mtDNA sequences and performed the initial set of experiments employing the 10X-based approach. E.K. performed all other experiments and data analysis. Z.J.G. provided expertise for MULTI-ATAC and 10X scATAC. E.K. and P.H.O.'F. wrote the manuscript with input from all authors.

## Competing interests

Z.J.G. is an author on a patent on MULTI-seq technology, and it has been licensed to Millipore. D.N.C. has consulted for MilliporeSigma about the MULTI-seq technology. The remaining authors declare no competing interests.
