## [Transparent Peer Review file · Nature Communications]

Selfish mutations promote age-associated erosion of mtDNA integrity in mammals

Corresponding Author: Dr Patrick O'Farrell

Version 0:

Reviewer comments:

Reviewer #1

(Remarks to the Author)

In their manuscript, Korotkevich et al employ recent advances in single-cell profiling of mitochondrial DNA (mtDNA), combined with insightful and relevant mathematical modelling, to describe and better understand accumulation of clonal and/or random mtDNA mutations over time.

While it would be interesting to see similar analysis in actual postmitotic cells with direct relevance for mitochondrial and other neurodegenerative conditions, they understandably focus on hepatocytes, supposedly because of their ease of dissociation, and high mtDNA copy number. Starting from wild-type aging mice, and a Polg mutator mouse model, they extend their findings to humans, by next also profiling hepatocytes from 6 human livers of various ages.

Their main findings are first a striking accumulation of random mtDNA mutations over time, with aging, with the existence of a few hot-spots for mutagenesis. While these findings are not novel, the scale of analysis is crucial, allowing modelling of many different mutations, and inferring underlying mechanisms controlling their emergence and accumulation. Second, using a clever approach, plotting average per-cell abundance of a single mutation against the number of cells affected, they distinguish driver from passenger mutations. This builds on older work, including from the same group, that specific mtDNA molecules can have a selfish (replicative?) advantage over others within the same cell.

This is a solid manuscript, well written, with clear and convincing figures, technically sound and exciting, using state-of-the-art approaches to study an important question in the field of mtDNA biology and disease, and advancing our thinking around these problems.

I am supportive of publication, but have a few, three slightly major and otherwise mostly minor, concerns that should be addressed first.

Major comments:

1. Line 69: the authors describe that they developed high-throughput sequencing methods. However, the approaches are very similar to what has been published over the last few years, in particular from the Sankaran, and later Ludwig labs. The authors should make it clear, in a separate paragraph either in the results or in the methods section, how their approaches differ from these published techniques, explaining the reason for the differences, and if any comparisons were done, showing data supporting this. This will add significant benefit to the field. In addition, I have a few specific questions that should be addressed:

- Line 648: Composition of permeabilization buffer: is this based on previously published work, and has optimisation been conducted?

- Ext Fig 2c: the percent of reads mapping to the mtDNA is substantially higher than previously published protocols. The authors should comment on this, and it would be interesting to see a comparison of these data to copy number measurements from single hepatocytes.

2. The model in Fig.2 does not take into account any cell division or early developmental/germline variants. While I understand the reasoning for this, and it is not required to reason conceptually about random/driver/passenger mutations, I do believe this should be taken into account when studying the human and mouse data. The authors should at least comment on this possibility, both in the results and in the discussion. In addition, I have a few specific points related to this:

- Line 154/Fig.3c and Ext Fig 5a, and Line 199: it seems most likely that the two purple OriL mutations are germline inherited variants that are present in their mouse strain, as they are detected in almost all of the cells across two separate experiments. It seems unlikely that shared variants found in such a large proportion of the cells could derive from multiple

independent de-novo mutation events in individual hepatocytes. An inherited germline (or early developmental) mutation would be a more plausible explanation and is supported by Ext Fig 3a, which shows the presence of these variants in the 3 month old WT mice.

- It might be helpful to separate some of the plots by animal in supplementary figures (eg. Fig. 3d, 4b, 4d, 4f), and provide a supplementary table with all the variants/abundances/cell numbers, and the mouse of origin in Fig. 3a and Ext. 5a.
- Fig. 3a: the same plot should be shown for the mutator mouse data, as well as for the aged human liver data (which seems missing from Ext fig 9?).
- Line 275: "instances" of passenger mutations linked to 3243A>G are mentioned, implying multiple different passengers were seen, but only one example is shown in 5i/h. The authors should in supplementary report the data for all identified passenger mutations, including how many cells each was found in. This will also give confidence that the various cells in Fig.5g are not clonally related (i.e. if they carry multiple independent passenger mutations and there are not multiple cells sharing a common passenger, which would instead suggest a clonal origin).

3. I could not find specification of the ethical framework for the use of the human data. This should be provided as well. In addition, I believe it would be helpful to have some clinical information about the individuals in a supplementary table, although this is not required if not available.

Minor comments:

- Line 38: this is an old reference from rats, which should either be specified, or another more relevant reference chosen (if any).
- Line 128: what is meant by fixation here? Do mutations indeed reach homoplasmy?
- Line 338: are the authors certain that the nuclear background of the mutator and the C57BL6J mouse is indeed different? In addition, I wonder whether they have the power to confidently say that this co-occurrence (or not) of driver mutations is not just by chance.
- Ext fig1c: could the coverage drop across the NCR be due to alignment to a linear genome, and could this be solved by shifting their start/end sites?
- Did the authors compare their NCR mutation rates to published mutation rates from other datasets (eg PMID: 38851187)? Could they also clarify the units being used to report mutation rates and make sure this is stated clearly and consistently in the main text and relevant figure legends? e.g. in line 120 the units are 'per base pair per replication cycle', in ext fig 7a legend 'per base pair per generation' and in ext fig 5b/c no units stated. NB: It may be more appropriate for the authors to use the 'per base pair per replication cycle' unit, as 'per base pair per generation' is usually used in literature to refer to generations at the organism level
- It would be helpful if the authors could comment on their definition (and cut-offs?) for driver mutations early in the manuscript.
- The ordering of the figure panels is not always logical (eg figure 1), and it might be helpful for the readers to consider this. It would also help to repeat the colour legend in Fig. 3. A schematic of the NCR zooms might help to interpret panels in Fig3. It might be useful to combine some extended data figures as well?
- The figure legends should be spell/grammar checked (eg last line of Fig.3 legend: passed/past; divers/drivers etc).

Reviewer #2

(Remarks to the Author)

Reviewer #3

(Remarks to the Author)

This manuscript by Korotkevich et al. explores how mtDNA mutations increase in abundance in mice and humans with age. Using single-cell mtDNA sequencing combined with computational stimulation, the authors detect an increase in abundance of mutant alleles. They rule out neutral drift as an explanation and instead suggest that positive selection underlies their rise. Authors also explore the possibility that many mutations are passenger mutations that rise in frequency by hitch-hiking along with driver mutations. Thus, affected genomes can rise in frequency along with associated passenger mutations, some of which may be detrimental. Ultimately, such "driver-passenger" associations represent an important mechanism by which de novo somatic mtDNA mutations can rise in abundance within cells and contribute to the age-associated accumulation of mtDNA mutations. As an example of this driver-passenger phenomenon, the authors characterize how the mtDNA-associated disease allele 3243A>G acts as a driver that can carry other mtDNA mutations to high cellular abundance.

Overall, the manuscript is well written, the methods are innovative, and the experiments are comprehensive. Notably, the authors leverage ATAC-seq coupled with FACS to detect de novo somatic mtDNA mutations, overcoming the limitations of standard bulk cell Illumina sequencing and greatly simplifying the interpretation of the data. The conclusions are well supported by the data. We have no major concerns with the manuscript. One suggestion is for the authors to consider that the key in Figure 1 be in other figures as well to increase readability.

[names redacted]

Version 1:

Reviewer comments:

Reviewer #1

(Remarks to the Author)

The authors have thoroughly addressed all our comments, both in the rebuttal and in the revised manuscript. I am very much looking forward to reading the more technical paper from Conrad et al., and am convinced by their reasoning and statistics around the mutation rates in the homopolymeric regions. Rebuttal figure 2 is really insightful in this respect, and would have been helpful in the manuscript as it takes away some of our confusion, but I agree that this only concerns a minor part of the manuscript, is discussed now in the results section, and understand that a more detailed analysis of these observations will be conducted separately.

This is a solid and important manuscript and I am supportive of publication, without any further concerns.

Reviewer #2

(Remarks to the Author)

RESPONSE TO REVIEWERS' COMMENTS

Reviewer #1

We appreciate the support for our manuscript and evident depth of technical considerations. We have made every effort to thoroughly address all comments and suggestions. However, we have refrained from making broad technical recommendations to the field, as we feel that any modifications that we introduced to improve our results are specific to our procedures and our cells and hence do not provide a foundation for generalization. Establishing such recommendations would require detailed comparisons across numerous experimental contexts, which would be more appropriate for a dedicated methods paper.

Major comments:

1. Line 69: the authors describe that they developed high-throughput sequencing methods. However, the approaches are very similar to what has been published over the last few years, in particular from the Sankaran, and later Ludwig labs. The authors should make it clear, in a separate paragraph either in the results or in the methods section, how their approaches differ from these published techniques, explaining the reason for the differences, and if any comparisons were done, showing data supporting this. This will add significant benefit to the field. In addition, I have a few specific questions that should be addressed:

Indeed, our approaches are very similar to those in previously published works from the Sankaran and Ludwig labs, which we cited in our manuscript. We apologize for not acknowledging the precedence of these papers and have modified the Results section to better reflect this. However, even though our efforts were behind these developments and in some cases were informed by them, we did not simply adopt those procedures: we developed our procedures largely independently and in parallel. We started by sequencing mtDNA from hand-dissected single muscle fibers, then moved on to developing a plate-based single-cell mtATAC (scmtATAC) approach building on a previously published bulk ATAC protocol (Buenrostro et al., 2013), and finally progressed to the establishment of the high-throughput approach based on a combination of the 10X-platform with MULTI-seq (Conrad et al., submitted).

As pointed out by the reviewer our approaches differ from previously published techniques in terms of permeabilization buffer (discussed below). We have not compared the performance of our buffer to that used by the Sankaran and Ludwig labs. In our hands, the described permeabilization buffer was effective when applied to mouse and human hepatocytes.

In addition to a different permeabilization buffer, our 10X-based approach is significantly enhanced by the new MULTI-seq method (Conrad et al., submitted). Combining 10X-scmtATAC with MULTI-seq not only allows the processing of multiple samples in a single experiment, minimizing technical variability and cost, but also enables the identification of

multiplets, which is critical for our study and virtually impossible to achieve with other methods, such as read-count-based AMULET method (Thibodeau et al., 2021), due to the variable ploidy of hepatocytes. We have added mention of these advantages to the Results and Methods sections.

- Line 648: Composition of permeabilization buffer: is this based on previously published work, and has optimisation been conducted?

We tested multiple permeabilization buffers for the effective generation of mtATAC libraries from mouse hepatocytes and arrived at a formulation different from that developed by the Sankaran and Ludwig labs. In our experience, adjustments to the permeabilization buffer (see Methods) were necessary when we moved from dissociated mouse hepatocytes to commercially available suspended human hepatocytes. As such, every cell type might have a different optimum, and hence, “optimization” is unlikely to yield a universally superior permeabilization buffer.

- Ext Fig 2c: the percent of reads mapping to the mtDNA is substantially higher than previously published protocols. The authors should comment on this, and it would be interesting to see a comparison of these data to copy number measurements from single hepatocytes.

Indeed, the percent of reads mapping to mtDNA in our study is higher than in previously published studies: ~75-95% vs ~20% (e.g., Lareau et al., 2021). This difference likely stems from vastly different mtDNA copy numbers in the analyzed cells and is not due to differences in the protocols. Previous studies focused on low-copy-number cells such as human hematopoietic cell lines and PBMCs (according to literature these cells have a few hundred copies of mtDNA), while we work with hepatocytes that have a very high mtDNA copy number (~10,000 copies per cell based on our ddPCR measurements). We have added a note on this point to the legend of Supplementary Fig. 2c.

It appears that the reviewer is asking whether read depth is correlated with mtDNA copy number for individual cells. We don't know how this can be done, as both ddPCR-based determination of copy number and mtDNA sequencing consume the sample. We do note, however, that the average read depth (100-500x) is well below the mtDNA copy number of mouse hepatocytes (~10,000).

2. The model in Fig.2 does not take into account any cell division or early developmental/germline variants. While I understand the reasoning for this, and it is not required to reason conceptually about random/driver/passenger mutations, I do believe this should be taken into account when studying the human and mouse data. The authors should at least comment on this possibility, both in the results and in the discussion. In addition, I have a few specific points related to this:

We note that mouse hepatocytes are largely quiescent – 95% of hepatocytes remain quiescent over 1.5 years (Arrojo e Drigo et al., 2019). Given that 1.5 years represents most of the life of an adult mouse, it is reasonable to model mouse hepatocytes as non-dividing cells for the purpose of exploring events happening in an adult. Moreover, modeling efforts by others have indicated only minor differences between homeostatic and mitotic turnover assumptions (An et al., 2024).

We agree with the reviewer that it is important to consider early-arising variants (germline variants and clonally amplified mutations that emerge during early development), which we did throughout our work. We have modified the Results section to reflect this more clearly. Early mutations contribute to many or all cells of the organism. Such events are not reflected in our models but can be identified in our datasets. Early mutations are rare; hence they affect only one or a few animals. Since all of the reported analyses examine multiple mice, early mutations can be readily identified by comparison of independent individuals. For example, the missense mutation marked with a black arrowhead in Supplementary Fig. 3 that is present in many cells of one 3-month-old WT mouse represents one such sporadic event. The most important implication of such early events for our findings is that, within a mouse harboring an early clonal mutation, numerous cells will have a substantial presence of this particular allele, which will result in frequent driver-passenger associations. Indeed, among the few passengers we identified in aged WT mice several were clonal mutations (see new Supplementary Fig. 9). This observation is relevant to inherited mtDNA disease alleles, which can similarly be captured by the somatic emergence of a driver allele and thereby driven to detrimental levels in somatic cells. We modified the Discussion section to reflect these points.

- Line 154/Fig.3c and Ext Fig 5a, and Line 199: it seems most likely that the two purple OriL mutations are germline inherited variants that are present in their mouse strain, as they are detected in almost all of the cells across two separate experiments. It seems unlikely that shared variants found in such a large proportion of the cells could derive from multiple independent de-novo mutation events in individual hepatocytes. An inherited germline (or early developmental) mutation would be a more plausible explanation and is supported by Ext Fig 3a, which shows the presence of these variants in the 3 month old WT mice.

The reviewer suggests that “the two purple OriL mutations are germline inherited variants that are present in their [our] mouse strain”. We appreciate this point but respectfully argue, based on multiple observations and the known behaviors of indels in homopolymers described below, that these mutations arise somatically and attain high frequency due to high mutation rate.

1. OriL mutations are detected in a mouse line with mtDNA unrelated to that of the C57BL6 strain.

In a control experiment presented in Supplementary Fig. 2, we analyzed mutations accumulating in hepatocytes of the conplastic mtPWD strain. These mice were produced by

Figure 1. AAA vs C# plot for hepatocytes from 22-month-old mtPWD mouse.

introducing mtDNA from genetically distant PWD/Ph mice into C57BL6 mice (Gregorová et al., 2008). As such, mtPWD and C57BL6 mice harbor mtDNA from distinct, unrelated sources. As shown in Response Fig. 1, the two OriL alleles (+A and -A) that we detect in all WT C57BL6/J and mutator mice are also seen in this mouse at similarly high frequencies. This is not consistent with the suggestion that they are heritable alleles carried only in our mice.

2. OriL mutations occur in a long homopolymer expected to have high rate of indels.

OriL sequence includes a stretch of 11 As, and we detect not two, but four indel alleles in this stretch: +/-A are the most prevalent, followed by +/-AA. Occasionally we also observed +AAA allele.

Homopolymeric sequences are well-documented mutational hotspots characterized by high frequencies of indels driven by polymerase slippage (Tran et al., 1997; Longley et al., 2001; Moran, McLaughlin and Sorek, 2009). It is known that indel frequency increases dramatically with homopolymer length. It is also known that single-nucleotide alterations in length are more frequent than changes of two, which are more frequent than changes of three.

As presented in the manuscript, for somatically emerging mutations, analysis of the distribution of allele abundance among cells can distinguish the contributions of mutation rate and selection. Our simulations show that alleles with increasing mutation rate follow a distinct rising path on an AAA vs C# plot (Supplementary Fig. 5b,c). Given what is known about slippage mutations in homopolymeric sequences and the prevalence of such sequences in mouse mtDNA (one stretch of 11 As, two stretches of 8 As, six stretches of 7 As and one stretch of 7 Cs), we expect indels in long homopolymers to be a major class of alleles following this path. Furthermore, their position along this path ought to be related to the length of the homopolymer and whether the particular allele involves changes of a single nucleotide or more. Response Fig. 2 (top panel) shows an annotated AAA vs C# plot for three 24-month-old WT mice. Alleles within the three homopolymeric runs, which are among the longest in mtDNA, are highlighted, and their mutation rate and selection coefficient estimates are shown. What can be observed is that these three regions make predominant contributions to mutations positioned in the area predicted for alleles with high mutation

Figure 2. AAA vs C# plot for hepatocytes for three 24-month-old WT mice. Top plot shows aggregated data for three mice. Bottom plots show data for each mouse separately. To estimate selection coefficient (s) and mutation rate (ϵ) parameters, observed data were tested against simulation outputs. Since back mutations become impactful for sites with high mutation rate our model included back mutation.

rate. Furthermore, the specific alleles are ordered such that those in the longer homopolymers show a higher mutation rate (located further to the right and higher up). Finally, within a homopolymer, alleles are ordered such that single-nucleotide-indels are more frequent than indels of two nucleotides. These specific allele behaviors are reproduced across mice, as shown by the plots for the individual mice in Response Fig. 2 (bottom panels).

The mutation rates estimated for the OriL +A and -A (8×10^{-3} and 2×10^{-3} , respectively) are high, especially when compared to the estimated SNP mutation rate (2×10^{-8}). It should be appreciated that the given mutation rates are for the described change, but there are 11 ways to add or remove one base pair (bp) in 11-mer. Thus, for comparison to SNP rates, these rates should be converted to a per bp numbers: 9×10^{-4} and 2×10^{-4} . While we do not have precise independent data on indel rates for homopolymers in mouse mtDNA, to estimate a plausible range of rates, we can turn to the characterization of such mutations in other systems. For example, Tran et al., 1997 provided a detailed assessment of how homopolymer features and repair functions influence indel frequencies in the yeast nuclear genome. Tran's measurement of mutation rates in homopolymers varied from 5.4×10^{-9} for a run of 4 As to 1.64×10^{-7} for a run of 14 As. Strikingly, impairing mismatch repair led to a dramatic increase of the rates: 4.6×10^{-8} and 1.6×10^{-3} , respectively. In another study of human mtDNA polymerase fidelity *in vitro*, the rate of single-nucleotide deletion for a run of 4 Ts was estimated at $< 10^{-5}$, while for a run of 8 Ts the estimate was a staggering 2×10^{-2} (Longley et al., 2001). Thus, our high mutation rate estimates for indels in OriL fall within the very broad range of estimates for indels in long homopolymers in other systems.

3. Distinguishing alleles with high mutation rate from clonal alleles.

The data presented in Response Fig. 2 show a clone (synonymous mutation) whose position on the AAA vs C# plot is close to that predicted for somatically emerging high-mutation-rate alleles. Such clones are sporadic. This one occurred in mouse #3 only. Its position on the AAA vs C# plot is in accord with the reviewer's suggestion that clonal expansion could have an appearance on these plots similar to that of high mutation frequency. However, the pattern of the high-mutation-rate alleles is reproduced in all mice that we have examined, whereas clonal alleles are sporadic both in their occurrence across mice and in their positions on AAA vs C# plots.

4. Considering a possibility of maternal transmission.

While the sporadic nature of clonal mutations allows us to identify them, as the reviewer points out, it might be possible that an abundantly represented mutation of maternal origin will be reliably transmitted and therefore misclassified. While the results we describe above are not attributable to a specific maternal strain, we consider here whether such transmission could plausibly explain the observed reproducible levels of the OriL alleles. Note that there are at least 5 alleles to consider (the WT, +A, -A, +AA, and -AA). All of these are found in the aged mice we examined. Thus, if they were of maternal origin, they would all have to be present in the mother and all successfully transmitted. Since the abundances of the different alleles are reproducible among different mice, maternal transmission would

have to involve accurate segregation (unknown for mtDNA) or a very large pool of genomes to ensure the maintenance of genomes with high (WT present at >80% in bulk), medium (+A and -A present at ~10% in bulk) and low (+AA and -AA present <1% in bulk) abundances without significant variance. Given what is known about bottlenecks in the transmission of mtDNA, the accurate maintenance of different abundance classes across all progeny is implausible. Furthermore, if we consider the entire group of reproducibly appearing indel alleles in homopolymeric sequences (Response Fig. 2) this becomes even more implausible. In contrast, the known high rate of indels in homopolymeric sequences accounts for the reproducible abundance distributions of these mutations in all mice.

These analyses support our conclusion that the OriL indels emerge somatically and climb to relatively high abundance in many cells due to their very high mutation rate. We would also like to emphasize that this is not a major point in the manuscript. Indeed, we intentionally set aside these alleles from detailed consideration to focus on what we believe is the new and novel feature of the paper – the report on the operation of selection and its role in the context of the driver-passenger relationship. We intend to pursue other findings, including the characterization of mutations in homopolymeric sequences, and report these separately.

- It might be helpful to separate some of the plots by animal in supplementary figures (eg. Fig. 3d, 4b, 4d, 4f), and provide a supplementary table with all the variants/abundances/cell numbers, and the mouse of origin in Fig. 3a and Ext. 5a.

We have provided abundance distributions of driver alleles separated by animals in a new Supplementary Fig. 7. We have reported all variants, their mean abundance in positive cells and number of positive cells for data presented in Fig. 3a and Supplementary Fig. 5a in a new Supplementary Table 2.

- Fig. 3a: the same plot should be shown for the mutator mouse data, as well as for the aged human liver data (which seems missing from Ext fig 9?).

We have added AAA vs C# plots for 24-month-old mutator mice (new Supplementary Fig. 6a) and an 81-year-old human (new Supplementary Fig. 9e).

- Line 275: “instances” of passenger mutations linked to 3243A>G are mentioned, implying multiple different passengers were seen, but only one example is shown in 5i/h. The authors should in supplementary report the data for all identified passenger mutations, including how many cells each was found in. This will also give confidence that the various cells in Fig.5g are not clonally related (i.e. if they carry multiple independent passenger mutations and there are not multiple cells sharing a common passenger, which would instead suggest a clonal origin).

We added all examples of passenger mutations linked to 3243A>G and their distribution among all sequenced cells of the individual to the new Supplementary Fig. 12.

3. I could not find specification of the ethical framework for the use of the human data. This should be provided as well. In addition, I believe it would be helpful to have some clinical information about the individuals in a supplementary table, although this is not required if not available.

Human samples were deidentified by the providers. We moved the sentence stating this to the beginning of the Methods section to make it more findable and added a statement that work with such samples is not considered human subject research. As requested, we added a table with all of the limited available clinical information about the analyzed human individuals (new Supplementary Table 3).

Minor comments:

- Line 38: this is an old reference from rats, which should either be specified, or another more relevant reference chosen (if any).

In the introduction we have replaced the reference to the study of mtDNA turnover in rats (Gross, Getz and Rabinowitz, 1969) with a recent human study (An et al., 2024). Of note, very few studies have measured mtDNA turnover directly. Attempts to measure mtDNA turnover in humans using heavy water (e.g., Collins et al., 2003) are likely to be inadequate since labeling is indirect and thus the results likely represent the complexity of H₂O metabolism rather than providing a measure of mtDNA turnover. Estimates of mtDNA turnover in humans based on mutational spectra (10-20 replacements per year; An et al., 2024), which we now cite in the introduction, closely agree with the direct data from rats that we relied on in our work (6-27 replacements per year; Gross, Getz and Rabinowitz, 1969).

- Line 128: what is meant by fixation here? Do mutations indeed reach homoplasmy?

By “fixation” we mean that a mutation reaches a state of homoplasmy in the analyzed/simulated cell, i.e., all genomes in the cell contain the mutation. A short explanation was added to the Results section.

- Line 338: are the authors certain that the nuclear background of the mutator and the C57BL6J mouse is indeed different? In addition, I wonder whether they have the power to confidently say that this co-occurrence (or not) of driver mutations is not just by chance.

The mutator mouse was created using 129Sv AB2.2 ES cells. The resulting animals were backcrossed to BL6 mice. However, even after many generations, we expect that genes around the mutant Polg sequence remain of 129Sv origin.

mtDNA competition is sensitive to gene expression (Samuels *et al.*, 2013) and nuclear genetic background (Gupta *et al.*, 2023). Even changes in gene dose have a large impact (Chiang *et al.*, 2019). Of particular relevance, reducing the dose of Polg in *Drosophila* entirely

reversed the competition between genomes such that a previous loser genome completely displaced the prior winner. We are certain of one major difference between the two mouse strains we analyzed: the sequence of Polg (WT vs mutant exo domain). It is likely that the Polg variant is a major determinant of which mtDNA alleles have an advantage in these two strains, but additional polymorphisms in the nuclear genome could also play a role.

The increased mutation rate in mutator mice allowed us to sample over a million mutations. The depth of this analysis ensures that our sensitivity for the detection of drivers in the mutator far exceeds that for the detection of drivers in WT. Thus, we can be confident that mutations detected as drivers in WT but not observed in the mutator are not drivers in the mutator. However, we cannot be confident the other way around, as it is possible that mutations that are drivers in mutator are also drivers in WT but are so infrequent that they did not pass our conservative thresholds. Indeed, we observed 16286A>G and 16293T>C at high levels in WT mice, suggesting that they are drivers that missed the thresholds. This was added to the legend of Fig. 3j.

- Ext fig1c: could the coverage drop across the NCR be due to alignment to a linear genome, and could this be solved by shifting their start/end sites?

Aligning reads with Gmap, which is aware that mtDNA is circular, or with Bwa using a reference with a shifted breakpoint of the genome indeed improves coverage at the very edge of the genome (positions ~16250 – 80; Response Fig. 3, arrowhead). While this improvement is significant (~2-fold increase in coverage in the middle of this region), it does not appreciably affect the estimates of levels of mutations present at high intracellular abundances. Interestingly, realigning sequencing reads with Gmap did not mitigate the large drop in coverage observed between 16100 and 16170 positions, which stretches from CSB2 up to the light strand promoter (Response Fig. 3, asterisk). A possible reason for this drop is a poor accessibility at this region due to binding by the replication/transcription complexes (Isaac *et al.*, 2024). A note on this has been added to the legend of Supplementary Fig. 1c.

Figure 3. Aligner effect on mtDNA coverage. Sequencing reads from the plate-based mtATAC experiment presented in Supplementary Fig. 1c were aligned with Bwa-mem using the standard mouse mtDNA reference (blue trace), with Bwa-mem using a reference with the genome breakpoint (black arrowhead) shifted to position 7060 (green trace), and with Gmap (red trace).

- Did the authors compare their NCR mutation rates to published mutation rates from other datasets (eg PMID: 38851187)?

The mutation rate reported in PMID: 38851187 (Árnadóttir *et al.*, 2024) is not directly related to the mutation rate we report in our manuscript. The mutation rate in our report represents the rate of mutation emergence, whereas the authors of PMID: 38851187 report the number of mutations per number of opportunities, n ($n = I \times T$, where I is the number of genome sites and T is the number of transmissions or years). While such an analysis provides estimates of the mutation emergence rate for nuclear genes, it is not a direct estimate of mutation emergence in mtDNA. Instead, it is rather a measure of the combined action of mutation emergence, random events and selection. To elaborate on this point, for a newly emerged mutation in a germ cell to reach detectable levels, it must successfully compete with co-resident genomes. The vast majority of neutral mutations are lost in this process and thus are never detected, resulting in an underestimation of the neutral mutation rate. However, mutations that are positively selected in the germline and/or in soma can reach very high levels and be readily detectable, thereby providing overestimated mutation rates. Of note, the authors of PMID: 38851187 report that a number of sites in the D-loop and position 3243 are “hypermutable”; however, they do not provide an explanation for this exceptional behavior. We found that mutations at several of these positions are under positive selection (152, 16093 and 3243) and it is likely that the rest of the hypermutable D-loop variants are also drivers either in the germline or in the tissues analyzed in PMID: 38851187.

Our mutation rate estimates for the driver mutations in the D-loop ($10^{-8} - 3.16 \times 10^{-7}$ per bp per replication cycle) are close to previously reported mtDNA mutation rates determined in the human B cell line TK6 (2×10^{-8} per bp per cell division; Khrapko *et al.*, 1997), human colon (5×10^{-5} mutations per genome per day which translates into 2.5×10^{-8} per bp per cell division assuming crypt stem cells divide once per 3 days; Taylor *et al.*, 2003) and clones from human colorectal epithelium, hematopoietic stem cells and fibroblasts (5×10^{-8} per base pair; An *et al.*, 2024). We have added a statement on this to the Results section.

Could they also clarify the units being used to report mutation rates and make sure this is stated clearly and consistently in the main text and relevant figure legends? e.g. in line 120 the units are ‘per base pair per replication cycle’, in ext fig 7a legend ‘per base pair per generation’ and in ext fig 5b/c no units stated. NB: It may be more appropriate for the authors to use the ‘per base pair per replication cycle’ unit, as ‘per base pair per generation’ is usually used in literature to refer to generations at the organism level

We thank the reviewer for pointing out the inconsistency. When discussing data, we have changed references to mutation rates to per base pair per replication cycle throughout the manuscript. We define Generation (with a capital G) as term referring to a cycle of mtDNA replacement (twice the half-life) in our simulations. The capitalization is intended to distinguish this use from more conventional uses of the word, such as generations of mice.

- It would be helpful if the authors could comment on their definition (and cut-offs?) for driver mutations early in the manuscript.

Cut-offs for driver mutations have been added to the Results section.

- The ordering of the figure panels is not always logical (eg figure 1), and it might be helpful for the readers to consider this. It would also help to repeat the colour legend in Fig. 3. A schematic of the NCR zooms might help to interpret panels in Fig3. It might be useful to combine some extended data figures as well?

Labeling (a through f) in Fig.1 is in accord with mentions of panels in the text. The order in the figure is odd as we tried our best to minimize the space and align panels c and f for easy comparison. We repeated the color legend as requested in all of the relevant figures and added a schematic representing zooming into NCR.

- The figure legends should be spell/grammar checked (eg last line of Fig.3 legend: passed/past; divers/drivers etc).

We thank the reviewer for pointing out these typos. These errors have been corrected. And we proofread all other figure legends.

Reviewers #2 and #3

We thank **[names redacted]** for thoughtful review of our manuscript and sincerely appreciate their positive feedback.

One suggestion is for the authors to consider that the key in Figure 1 be in other figures as well to increase readability.

We thank the reviewers for pointing out this shortcoming. We have added the legend for all applicable figures.

References

An, J. *et al.* (2024) 'Mitochondrial DNA mosaicism in normal human somatic cells', *Nature Genetics*. Springer US, 56(8), pp. 1665–1677. doi: 10.1038/s41588-024-01838-z.

Árnadóttir, E. R. *et al.* (2024) 'The rate and nature of mitochondrial DNA mutations in human pedigrees', *Cell*, 187(15), pp. 3904-3918.e8. doi: 10.1016/j.cell.2024.05.022.

Arrojo e Drigo, R. *et al.* (2019) 'Age Mosaicism across Multiple Scales in Adult Tissues', *Cell Metabolism*. Elsevier Inc., 30(2), pp. 343-351.e3. doi: 10.1016/j.cmet.2019.05.010.

Buenrostro, J. D. *et al.* (2013) 'Transposition of native chromatin for fast and sensitive epigenomic profiling of open chromatin, DNA-binding proteins and nucleosome position', *Nature Methods*, 10(12), pp. 1213–1218. doi: 10.1038/nmeth.2688.

- Chiang, A. C. Y. *et al.* (2019) 'A Genome-wide Screen Reveals that Reducing Mitochondrial DNA Polymerase Can Promote Elimination of Deleterious Mitochondrial Mutations', *Current Biology*. Elsevier Ltd., 29(24), pp. 4330–4336.e3. doi: 10.1016/j.cub.2019.10.060.
- Collins, M. L. *et al.* (2003) 'Measurement of mitochondrial DNA synthesis in vivo using a stable isotope-mass spectrometric technique', *Journal of Applied Physiology*, 94(6), pp. 2203–2211. doi: 10.1152/jappphysiol.00691.2002.
- Gregorová, S. *et al.* (2008) 'Mouse consomic strains: Exploiting genetic divergence between *Mus m. musculus* and *Mus m. domesticus* subspecies', *Genome Research*, 18(3), pp. 509–515. doi: 10.1101/gr.7160508.
- Gross, N. J., Getz, G. S. and Rabinowitz, M. (1969) 'Apparent turnover of mitochondrial deoxyribonucleic acid and mitochondrial phospholipids in the tissues of the rat.', *Journal of Biological Chemistry*, 244(6), pp. 1552–1562.
- Gupta, R. *et al.* (2023) 'Nuclear genetic control of mtDNA copy number and heteroplasmy in humans', *Nature*. Springer US, 620(7975), pp. 839–848. doi: 10.1038/s41586-023-06426-5.
- Isaac, R. S. *et al.* (2024) 'Single-nucleoid architecture reveals heterogeneous packaging of mitochondrial DNA', *Nature Structural and Molecular Biology*. Springer US, 31(3), pp. 568–577. doi: 10.1038/s41594-024-01225-6.
- Khrapko, K. *et al.* (1997) 'Mitochondrial mutational spectra in human cells and tissues', *Proceedings of the National Academy of Sciences of the United States of America*, 94(25), pp. 13798–13803. doi: 10.1073/pnas.94.25.13798.
- Lareau, C. A. *et al.* (2021) 'Massively parallel single-cell mitochondrial DNA genotyping and chromatin profiling', *Nature Biotechnology*. Springer US, 39(4), pp. 451–461. doi: 10.1038/s41587-020-0645-6.
- Longley, M. J. *et al.* (2001) 'The Fidelity of Human DNA Polymerase γ with and without Exonucleolytic Proofreading and the p55 Accessory Subunit', *Journal of Biological Chemistry*. © 2001 ASBMB. Currently published by Elsevier Inc; originally published by American Society for Biochemistry and Molecular Biology., 276(42), pp. 38555–38562. doi: 10.1074/jbc.M105230200.
- Moran, N. A., McLaughlin, H. J. and Sorek, R. (2009) 'The dynamics and time scale of ongoing genomic erosion in symbiotic bacteria', *Science*, 323(5912), pp. 379–382. doi: 10.1126/science.1167140.
- Samuels, D. C. *et al.* (2013) 'Recurrent Tissue-Specific mtDNA Mutations Are Common in Humans', *PLoS Genetics*, 9(11). doi: 10.1371/journal.pgen.1003929.
- Taylor, R. W. *et al.* (2003) 'Mitochondrial DNA mutations in human colonic crypt stem cells', *J Clin Invest*, 112(9), pp. 1351–1360. doi: 10.1172/JCI200319435.Introduction.
- Thibodeau, A. *et al.* (2021) 'AMULET: a novel read count-based method for effective multiplet detection from single nucleus ATAC-seq data', *Genome Biology*. Genome Biology, 22(1), pp. 1–19. doi: 10.1186/s13059-021-02469-x.

Tran, H. T. *et al.* (1997) 'Hypermutable Homonucleotide Runs in Mismatch Repair and DNA Polymerase Proofreading Yeast Mutants', *Molecular and Cellular Biology*, 17(5), pp. 2859–2865. doi: 10.1128/mcb.17.5.2859.